# Retrieval of daytime mesospheric ozone using OSIRIS observations of $O_2(a^1\Delta_g)$ emission

Anqi Li[1], Chris Roth[2], Kristell Pérot[1], Ole Martin Christensen[1], Adam Bourassa[2], Doug Degenstein[2], and Donal Murtagh[1]

[1]Chalmers university of technology, Department of Space, Earth and Environment, Gothenburg, Sweden
[2]Institute of Space and Atmospheric Studies, University of Saskatchewan, Saskatoon, Canada

**Correspondence:** Anqi Li (anqi.li@chalmers.se)

**Abstract.** Improving knowledge of the ozone global distributions in the mesosphere-lower thermosphere (MLT) is a crucial step in understanding the behaviour of the middle atmosphere. However, the concentration of ozone under sunlit conditions in the MLT is often so low that its measurement requires instruments with very high sensitivity. Fortunately, the bright oxygen airglow can serve as a proxy to retrieve the daytime ozone density indirectly, due to the strong connection to ozone photolysis in the Hartley band. The OSIRIS IR imager (hereafter IRI), one of the instruments on the Odin satellite, routinely measures the oxygen infrared atmospheric band (IRA band) at $1.27\,\mu m$. In this paper, we will primarily focus on the detailed description of the steps done for retrieving the calibrated IRA band limb radiance (with <10% random error), the volume emission rate of $O_2(a^1\Delta_g)$ (with <25% random error) and, finally, the ozone number density (with <20% random error). This retrieval technique is applied to a one-year-sample from the IRI data set. The resulting product is a new ozone data set with very tight along-track sampling distance (<20 km). The feasibility of the retrieval technique is demonstrated by a comparison of coincident ozone measurements from other instruments aboard the same spacecraft, as well as zonal mean and monthly average comparisons between Odin-OSIRIS (both spectrograph and IRI), Odin-SMR and Envisat-MIPAS. We find that IRI appears to have a positive bias of up to 25% below $75\,km$, and up to 50% in some regions above. We attribute these differences to uncertainty in the IRI calibration as well as uncertainties in the photochemical constants. However, the IRI ozone data set is consistent with the compared data set in terms of the overall atmospheric distribution of ozone between 50 and $100\,km$. If the origin of the bias can be identified before processing the entire dataset, this will be corrected and noted in the dataset description. The retrieval technique described in this paper can be further applied to all the measurements made throughout the 19-year mission, leading to a new, long-term, high resolution ozone data set in the middle atmosphere.

## 1  Introduction

The distribution of ozone plays a key role in the middle atmosphere. It can influence the radiative budget, thus affecting temperature structures and dynamic flow patterns (Brasseur and Solomon, 2005). As such, attention has been drawn to the observations of ozone over the past decades. The existence of the stratospheric ozone layer, resulting from the absorption in the Herzberg continuum, was proposed early last century and subsequently confirmed (Chapman, 1930). Near the mesopause region, a secondary ozone layer, which is the result of radiation absorption in the Schumann-Runge continuum, was later detected (Hays and Roble, 1973). More recently, the existence of a tertiary ozone maximum was discovered by multiple measurements both from ground-based and satellite instruments (e.g., Marsh et al., 2001). The tertiary ozone maximum only occurs in winter, in the high-latitude middle mesosphere. The mechanism behind it is mainly due to a decrease in atomic oxygen losses involving the odd-hydrogen species near the polar night terminator. However, the detailed picture of the tertiary ozone maximum is not yet fully explained because of the complexities in both the chemical composition and the dynamics of the mesosphere and lower thermosphere (MLT) (e.g., Hartogh et al., 2004; Degenstein et al., 2005a; Sofieva et al., 2014; Smith et al., 2018).

Satellite observations provide us with valuable knowledge on the behaviour of atmospheric ozone. In the MLT region, various measurement techniques are employed to monitor the ozone distribution. For instance, there are observations of ozone absorption by using solar or stellar occultation (e.g. HALOE, ACE-FTS, SOFIE, GOMOS) (all acronyms are given in Table A2), of emission from thermally excited ozone (e.g. SABER at $9.6\,\mu m$, MIPAS, SMR) and of airglow emission (SME, SABER and SCIAMACHY at $1.27\,\mu m$, OSIRIS at $762\,nm$). Smith et al. (2013) have shown comparisons of ozone concentrations in the MLT region resulting from most of the above mentioned techniques. They have concluded from coincident profile comparisons that different measurement principles agree with each other reasonably well (better than 20% for the instruments considered here). However, they emphasise that the differences in local time sampling among the measurements impact the inferred global distribution in the MLT, for instance, the vertical structure and seasonal variations of ozone. Additionally, differences in measurement principle, sampling schedules, uncertainties in the calibration and band-passes of the instrument, and inaccurate pointing knowledge may also be factors contributing to the difference between these ozone observations.

For ozone measurements based on inference assuming photochemical equilibrium, the photochemical timescales of the airglow species can critically affect the inferred ozone distribution, especially of the species whose lifetimes are comparable to the transport timescales. The $1.27\,\mu m$ oxygen emission has a photochemical lifetime of about 74 minutes (Newman et al., 2000) which can influence the retrieved ozone quantity in two ways. One is the effect of advective and diffusive transport of the relevant species, and the other is the delay in reaching quasi-photochemical equilibrium after sunrise. Zhu et al. (2007) have evaluated the uncertainties in daytime ozone retrieved from $1.27\,\mu m$ emission due to the effect of tidal waves and the photochemical steady state assumption, by using a dynamical-photochemical coupled airglow model.

In this study, we will focus on the retrieval of the ozone data collected by instruments aboard the Odin satellite, primarily the OSIRIS IR imager (hereafter IRI). The Odin satellite is orbiting the Earth around 15 times per day since 2001 and is still fully functional (Murtagh et al., 2002). SMR (SubMillimeterwave Radiometer) and OSIRIS (Optical Spectrograph and Infrared

Imaging System) are the two main components on Odin. Both instruments measure various species closely related to middle
atmospheric ozone chemistry by observing the Earth's limb. OSIRIS, in fact, consists of two optically independent instruments:
the optical spectrograph (hereafter OS) and the infrared imager. IRI has three vertical imagers. Two of them measure the oxygen
infrared atmospheric band (IRA band) emissions centred at $1.27\,\mu m$, and the third one measures the OH Meinel band emission
centred at $1.53\,\mu m$. A more detailed description of IRI can be found in Sect. 2.1 as well as in Llewellyn et al. (2004). Data
collected by one of the oxygen IRA band imagers have been studied by Degenstein et al. (2004) to demonstrate a tomographic
retrieval technique to derive airglow volume emission rate and its comparison to non-tomographic retrieved volume emission
rate. Degenstein et al. (2005b) showed the potential of the IRI observations for estimating ozone depletion during a Solar
Proton Event. The observations of oxygen IRA band and the OH Meinel band together were used to study the mesospheric
tertiary ozone peak by Degenstein et al. (2005a). To our knowledge, there is no further investigation which deals with the data
set from the IRI instrument.

Our primary objective in this paper is to revisit the oxygen airglow measurements obtained from the IRI at $1.27\,\mu m$ and
demonstrate a retrieval scheme used to derive the volume emission rate as well as the ozone concentration in the MLT region
based on Bayesian estimation. In addition, we address the issue of the validity of the photochemical equilibrium assumption
near the local sunrise by using a novel treatment in the ozone retrieval. This ozone product will be a completely new data set
from the Odin mission and is complementary to the already existing ozone measurements since the signal strength in the MLT
region during daytime is often too low for the other instruments. In addition, this IRI ozone product has about 70 times higher
along-track sampling rate than the other ozone products thanks to the imaging technique.

To illustrate the performance of the retrieval technique, a small but representative sample (every 20th orbit) of the IRI mea-
surements collected from November 2007 to October 2008 has been processed. Our secondary focus is to demonstrate the
fidelity of the resulting new IRI ozone product, by a side-by-side comparison of monthly mean zonally averaged distribution
with other independent ozone measurements, namely OS, SMR and MIPAS ozone products. However, we would like to em-
phasise that this paper is not intended to be a full validation study. IRI, OS and SMR observe at the same geographical location
and time because they are on board the same platform, thus the bias due to the different sampling schedules mentioned in
Smith et al. (2013) is negligible. However, to give a more complete picture of the IRI data set, we also include MIPAS ozone
profiles in our comparison (although this reintroduces the issues with local time sampling). Even though biases are found (see
Sect. 3.4), we find that the data set can reproduce the general seasonal and latitudinal pattern of the ozone distribution, which
indicates that the presented IRI ozone retrieval scheme can be applied to the whole 19 years of the mission to date, opening
new opportunities to perform further scientific studies. We also for the first time show results from all three ozone data sets
collected by the Odin satellite, illustrating how they complement each other well, despite their intrinsically different underlying
physical bases in terms of measurement techniques. And thus, by adding Ozone retrieved from the IRI instrument to Odin's
repertoire we expand the possibility for future studies using data from this fruitful research satellite.

## 2 Theory and implementation

In this section, we will discuss the necessary steps to derive the calibrated limb radiance (in Sect. 2.1), then the volume emission rate of the oxygen IRA band (in Sect. 2.2) and, finally, the ozone number density profiles (in Sect. 2.3). The theoretical background, the implementation details and the intermediate results with their estimated random errors can be found in the corresponding subsections. At the end of this section, data availability at 80 km and the estimated systematic error sources are discussed.

### 2.1 Level 1 data – calibrated limb radiance data

The IRI instrument measures the oxygen IRA band with 10 μm wide filters centred at 1.273 μm and 1.263 μm (channels 2 and 3, respectively, in OSIRIS nomenclature) and the OH Meinel emissions with a 40 μm wide filter centred at 1.530 μm (channel 1) (Degenstein et al., 2004). All three of the single-lens IR imagers consist of a linear array of 128 InGaAs photodiodes (pixels). Each array is split into two sections: a masked off, permanently dark portion of approximately twenty pixels used for calibration, and the remaining pixels used for data collection. The optical portion of the IRI instrument was designed such that the angular spacing between photodiodes results in approximately 1 km separation between the tangent altitudes of the look vectors. Each image of the IRI system consists of a measurement of each of the 128 pixels. Images are taken approximately every two seconds with a one-second duration exposure time.

Like any photodetection system, the IRI must be calibrated to remove instrument dependent effects from the measurement and convert the digital count into calibrated radiance. This calibration process occurs in four steps: (1) dark current and electronic offset correction, (2) relative calibration of the pixel gain, (3) removal of stray light, and (4) absolute calibration. The calibration process applied to the IRI data used in this work is an updated version of Bourassa (2003). A short description of each step follows.

In this paper, we will only look at data taken from channel 3 centred at 1.263 μm.

### 2.1.1 Dark Current and Electronic Offset

Each of the 128 pixels in the linear array of photodiodes has a unique temperature dependent dark current characteristic. The signal is referred to as "dark current" as it is thermally generated and present regardless of whether or not the photodiode is subject to light (photons). As is typical of semiconductor systems this small number of thermally generated electron-hole pairs has a Poisson distribution and follow the Shockley equation.

The electron-hole diffusion current and recombination current are proportional to

$$e^{-E_g/k_B T} \tag{1}$$

and

$$e^{-E_g/2k_B T}, \tag{2}$$

respectively, where $E_g$ is the band gap energy, $T$, the temperature, and $k_B$, Boltzmann's constant.

In practice, each pixel's unwanted thermal signal can be characterised by a single exponential term of the form

$$\gamma_i e^{\beta_i/T}, \tag{3}$$

for each pixel $i$ in the array, where $\gamma$ and $\beta$ are parameters found by implementing least-squares curve fitting to the data.

In addition to the removal of the dark current, two sources of electronic offset must also be characterised and removed from the measurements. The first is a relatively time-invariant electronic offset that is unique to each pixel. By adding a parameter that characterises each pixel's unique electronic offset to the above equation, a three-parameter fit is used for each pixel

$$\alpha_i + \gamma_i e^{\beta_i/T} \tag{4}$$

where $\alpha$ is the offset parameter. The second form of electronic offset is the same for each pixel, but varies randomly with each image due to noise in the electronics. This is handled separately from the three-parameter fit.

Calibration data for the IRI instrument, where an optical shutter is closed to block out incoming light, is used to compute the three parameters (for each pixel) at regular intervals throughout the mission. The fitting process is a periodised, least-squares optimisation.

By applying the parameters found using the calibration data to the data collection portion of the mission where the optical shutter is open, the dark current and the pixel dependent electronic offset are removed from the raw data. The image dependent electronic offset can then be determined, and subtracted off, using the permanently dark masked off pixels.

In short, this step calibrates each photodiode's measurement to zero (within measurement error consistent with shot noise) when not exposed to light.

### 2.1.2   Relative Calibration of the Pixels

In this step, referred to as the relative calibration or pixel "flat fielding", each pixel's output is normalised so that a uniform input brightness on each pixel results in the same digital counts. Prior to launch, the instrument was subjected to a calibrated Lambertian light source to determine these parameters, but early mission data revealed that the pre-launch relative calibration curves were no longer accurate.

To perform an in-flight relative calibration of the pixels, the mesospheric night-time airglow layer was used in place of a calibrated Lambertian source. As the IRI instrument scans up and down through this layer, comparisons are made between neighbouring pixels as they pass through the same layer to derive this relative gain factor for each pixel. Although the airglow layer is not constant in brightness, the statistical impact of this variation becomes negligible as the number of intercomparisons becomes large. The relative calibration algorithm was applied to every applicable night time orbit. The resulting data was averaged to create an in-flight relative calibration curve that is applied to the IRI data. The in-flight curves closely resemble the pre-flight curves with notable differences towards the edges of the arrays.

### 2.1.3 Stray Light Removal

It is evident from the IRI data that off-axis light from the sunlit Earth is incident on the IR detectors due to scattering and diffraction. An in-depth modelling of the IRI optical system was performed by Ivanov (2000). This work and Bourassa (2003) finds that during the sunlit portions of the orbit, the measured signal is the sum of the atmospheric brightness and a large unwanted stray light signal from the off-axis Earth below.

To remove the stray light, its shape is first characterised using data where the amount of real incident light is negligible. This occurs when the pixel look direction tangent points are over 100 km. The shape of the stray light is then extrapolated to lower tangent altitudes and the magnitude of stray light for any image is assumed to be proportional to the average brightness of the pixels over 100 km. As the number of pixels over 100 km changes from image to image due to the nodding nature of the Odin spacecraft, the quality of the stray light removal process changes; becoming less accurate when fewer pixels are present over 100 km. This decrease in accuracy is accounted for in the error estimate of the data related to the stray light removal process.

### 2.1.4 Absolute Calibration

Finally, the data is multiplied by a factor to convert the digital number measurement of the read-out electronics to a measurement of calibrated radiance reported in $\frac{\text{photons}}{\text{s} \cdot \text{cm}^2 \cdot \text{sterad}}$, or photons per second passing through a unit area within a unit solid angle.

The absolute calibration value used to convert the data from digital counts to brightness was determined through calibration sessions pre-launch. Post-launch the calibration value for the 1.53 μm channel was checked by comparing it to a standard single Rayleigh scattering model of the atmosphere, which showed that the pre-launch value was still applicable. As there is no equivalent simple atmospheric model to test against the 1.27 μm, and there has been no evidence to conclude otherwise, the assumption is that the absolute calibration values for channels 2 and 3 are also still applicable.

### 2.1.5 Calibration Error

Throughout each step in the calibration process, uncertainties are calculated so that the uncertainty values given with the final calibrated data are meaningful and accurate.

The error in the measured digital number as reported by the read-out electronics is a combination of two sources: the shot noise of the detector and the random error due to the fact that the number of photons incident on the pixel array follows a Poisson distribution (which is negligibly small for all but the brightest of scenes.)

The final reported error also incorporates the error in the pixel electronics offset, and is the combination of the errors determined through the various calibration steps: the error in pixel electronics offset and thermal characteristics, the error in the relative calibration curve, and finally the error in the stray light calibration table.

Figure 1 shows the radiance profile of a sample IRI exposure. The vertical axis is tangent height of the pixel's look direction rather than pixel number. For daytime exposures, a total calibration random error between 1-10% is typical with the stray light error as the largest contributing factor. Figure 2 shows a sample orbit of IRI limb radiance and its percentage error.

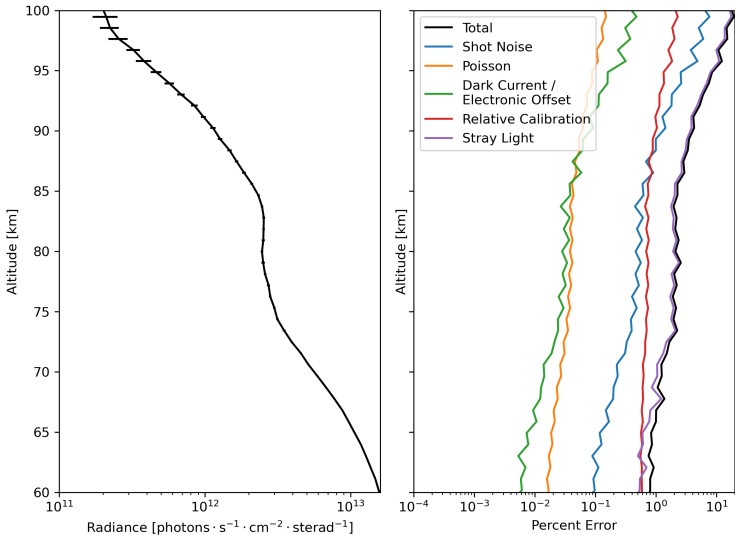

**Figure 1.** Left: A sample daytime exposure of IRI radiance data (image 1250 of orbit 37400, see Fig. 4 for reference). Right: The corresponding total calibration random error and its contributions.

## 2.2 Retrieval of $O_2(a^1\Delta_g)$ airglow volume emission rate

Once we have obtained the calibrated limb radiance, the next quantity to derive is the volume emission rate, since the volume
emission rate of photons emitted in the oxygen infrared atmospheric band is directly related to the number density of $O_2(a^1\Delta_g)$
by its radiative lifetime (i.e. Einstein A coefficient). In this paper, only daytime measurements are considered since we rely on
a photochemical scheme, described in Sect. 2.3, to derive the ozone number density. However, IRI also collects high quality
data in the night part of the orbits which is valuable for other studies.

The IR imager measures the limb radiance $R$ which is described by the radiative transfer equation

$$R = \frac{1}{4\pi} \int \phi V(s) e^{-\tau(s)} \mathrm{d}s, \tag{5}$$

where $V(s)$ is the volume emission rate over the full band at location $s$ along the line-of-sight of the instrument. $\phi$ is the
'filter factor' defined as the overlap between the instrument filter and the oxygen IRA band emission lines, and $e^{-\tau(s)}$ is the
transmissivity between the emission source at $s$ and the instrument along the line-of-sight. The value of $\phi$ is estimated from a
simple spectral calculation using the HITRAN (Gordon et al., 2017) catalogue for the emission line strengths. For simplicity,
we have estimated $\phi$ at the temperature of the tangent point since the signal is dominated by this emission.

In this paper, we will use a linearised scheme to retrieve volume emission rate profiles. The retrieval problem becomes linear
if we assume that the majority of the signal originates from the tangent layer emissions. However, absorption and scattering
processes may become important where the atmospheric layer at the line-of-sight is optically thick. In the case of the oxygen
IRA band emission, Degenstein (1999) has indicated that self-absorption is important to consider when line-of-sight tangent is

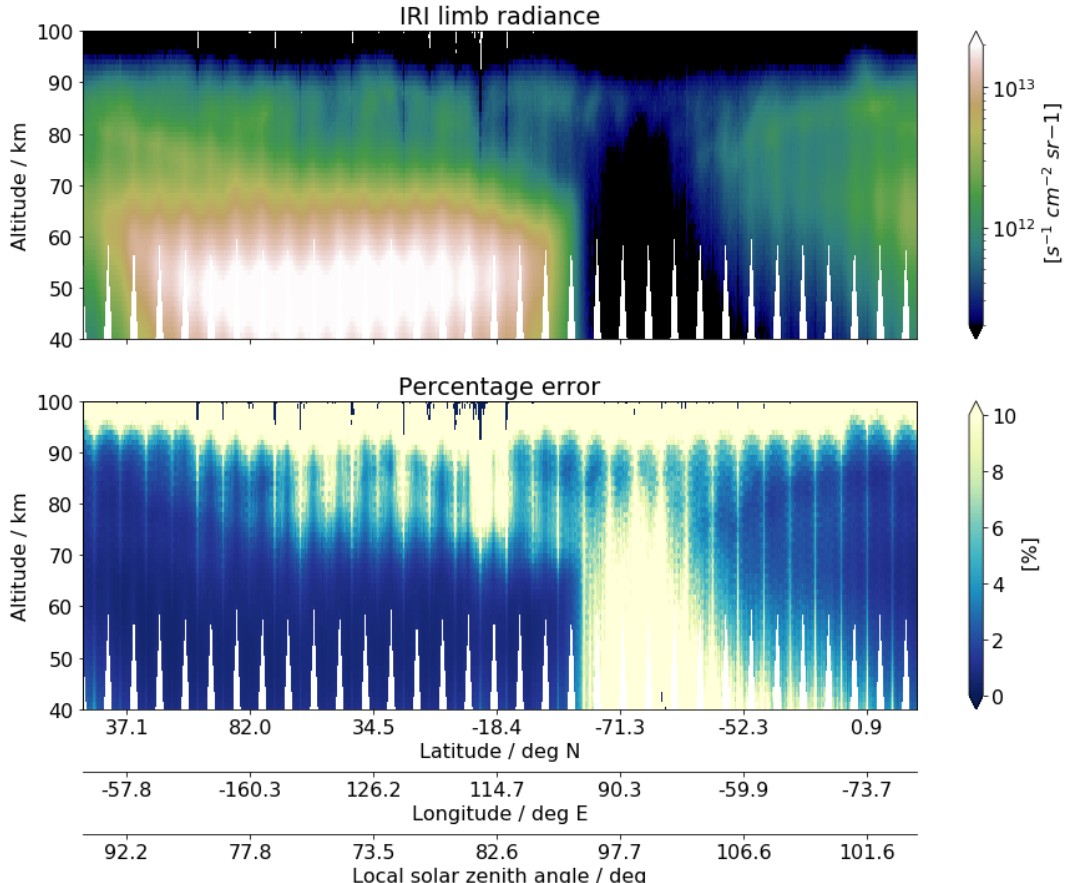

**Figure 2.** An example of IRI limb radiance data and its percentage error for one orbit collected on 2008-03-30, from 22:21:09 to 23:09:13 (orbit number 38720), as a function of geographical location and tangent altitude. Local solar zenith angle is also shown in the horizontal axis.

lower than $60\,\mathrm{km}$. In order to account for the issue of absorption, we pre-compute a table of the absorption factors $e^{-\tau(s)}$ as a function of the line-of-sight along the tangent path. The absorption coefficients are taken from HITRAN and MSIS (Picone et al., 2002) is used for temperature, pressure and $O_2$ density. The relationship between the measured limb radiance and volume emission rate can then be expressed as

$$y_i = \sum_j k_{ij} x_j + \epsilon_i = \frac{4\pi R_i}{\phi}, \tag{6}$$

where $y_i$ is the column emission rate for the full oxygen IRA band as would be measured by the pixel $i$ (which is proportional to $R_i$), $x_j$ the volume emission rate at the atmospheric layer $j$, $k_{ij}$ the path length of the line-of-sight $i$ through the atmospheric layer $j$ weighted by the absorption factor and $\epsilon_i$ the measurement errors. In matrix notation, the relationship can thus be

expressed as

$$\mathbf{y} = \mathbf{Kx} + \epsilon, \tag{7}$$

where $\mathbf{y}$ is also termed the measurement vector, $\mathbf{K}$ the weighting function, or Jacobian matrix and $\mathbf{x}$ the state vector.

In this paper, the optimal estimation method (OEM), also known as the maximum a posteriori (MAP) method (Rodgers, 2000), is employed to invert the above equation. By constraining the inversion using the uncertainties of both the measured quantity and the a priori knowledge, the estimated profile of the volume emission rate can be expressed as

$$\hat{\mathbf{x}} = \mathbf{x_a} + \mathbf{G}(\mathbf{y} - \mathbf{Kx_a}), \tag{8}$$

where $\mathbf{x_a}$ denotes the a priori profile of volume emission rate, and $\mathbf{G}$ the gain matrix, which is equal to:

$$\mathbf{G} = (\mathbf{K^T S_e^{-1} K} + \mathbf{S_a^{-1}})^{-1} \mathbf{K^T S_e^{-1}}, \tag{9}$$

where $\mathbf{S_e}$ and $\mathbf{S_a}$ are the error covariance matrices describing the uncertainties of the measurement $\mathbf{y}$ and of the a priori profile $\mathbf{x_a}$, respectively.

In our implementation, $\mathbf{x_a}$ is the $O_2(a^1\Delta_g)$ volume emission rate profile calculated by the photochemical model (see Sect. 2.3) by inputting the ozone profile from a climatology. This climatology was derived from the data presented by the Canadian Centre for Climate Modelling and Analysis known as the CMAM model [1] and evaluated for different latitudes, months and local solar times for the tangent points of the IRI measurements. The covariance matrix of the a priori follows

$$S_a(i,j) = \sigma_a(i)\,\sigma_a(j)\,\exp(|i-j|\frac{dz}{h}) \tag{10}$$

where $\sigma_\mathbf{a}$ is set to be $0.75\mathbf{x_a}$ and $dz/h = 1/5$. The off-diagonal elements act as a regularization on the estimation to prevent oscillations. $\mathbf{S_e}$ has diagonal elements equal to the square of the uncertainty of each pixel (i.e. the calibration random error described in Sect. 2.1). All off-diagonal elements for $\mathbf{S_e}$ are set to zero, that is assuming no correlation between errors for different pixels in the limb radiance measurements. The retrieval grid covers altitudes from $10\,\mathrm{km}$ to $130\,\mathrm{km}$ with $1\,\mathrm{km}$ spacing. To select the limb radiance measurement, a lower bound of $40\,\mathrm{km}$ and an upper bound of $100\,\mathrm{km}$ line-of-sight tangent height are chosen. Thus, a $30\,\mathrm{km}$ margin for both the lower and upper bounds in order to minimise any edge effect in the inversion process.

The vertical resolution of the retrieved data can be represented by the averaging kernel (AVK) matrix,

$$\mathbf{A} \equiv \frac{\partial \hat{\mathbf{x}}}{\partial \mathbf{x}} = \mathbf{GK} \tag{11}$$

which maps the changes from the true state $\mathbf{x}$ to the estimated state $\hat{\mathbf{x}}$ at corresponding altitudes. The sum of each row of AVK matrix is termed the measurement response (MR) which describes how sensitive the estimated state is to true atmospheric state. However, it is more convenient, here, to assess AVK and MR relative to the a priori profile. This is because $\mathbf{x_a}$ exhibits a strong

[1]CMAM data is downloaded at http://climate-modelling.canada.ca/climatemodeldata/cmam/cmam30/

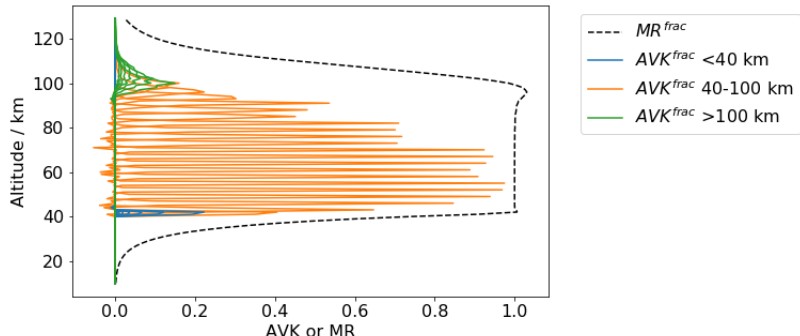

**Figure 3.** Every third row of the fractional AVK (solid lines) and measurement response (black dashed line) from the retrieval of volume emission rate of the image 570 indicated in Fig. 4.

vertical gradient and its covariance is scaled with $\mathbf{x_a}$ itself. As discussed in e.g. Baron et al. (2002); Hoffmann et al. (2011), the transformation from the ordinary AVK to the 'fractional AVK' matrix is given by,

$$A_{ij}^{frac} = x_a(j) \cdot A_{ij}/x_a(i). \tag{12}$$

Accordingly, the 'fractional MR' is given by,

$$MR_i^{frac} = \sum_j A_{ij}^{frac}. \tag{13}$$

An example of the rows of the fractional AVK matrix and the corresponding MR of an inversion is shown in Fig. 3. As we can see these curves generally peak at their corresponding altitudes between $40\,\mathrm{km}$ and $100\,\mathrm{km}$ where the line-of-sight tangent of the measurements lies. However, AVKs that represent $\hat{\mathbf{x}}$ above $100\,\mathrm{km}$ peak mostly around $100\,\mathrm{km}$ and their full width at half maximum (FWHM) become much larger. This indicates that the vertical resolutions of these altitudes are lower, which is a direct result of having no measurements at tangent altitudes above $100\,\mathrm{km}$. The retrieval resolution is about 1-2 km below $90\,\mathrm{km}$ altitude. Figure 3 also shows the fractional MR. It has a value close to unity between 40 and $100\,\mathrm{km}$ and quickly returns back to zero where no measurements are available. This indicates that the a priori profile has little influence on the estimated result between these altitudes. We use a fractional MR of 0.8 as the threshold to evaluate the quality of the estimated volume emission rate to present the remaining results and perform further analysis in this paper.

Besides MR, OEM also provides us with an analytical expression of the uncertainty in the estimated quantity. The covariance of retrieval noise is

$$\mathbf{S_m} = \mathbf{G S_e G^T}. \tag{14}$$

The diagonal elements of $\mathbf{S_m}$ will be treated as $\mathbf{S_e}$ in Sect. 2.3.

Figure 4 displays a typical example of the estimated volume emission rate and the random error (i.e. $\mathbf{S_m}$) relative to the a priori, retrieved along one orbit. Only daytime measurements with a fractional MR greater than 0.8 are shown here. Two airglow

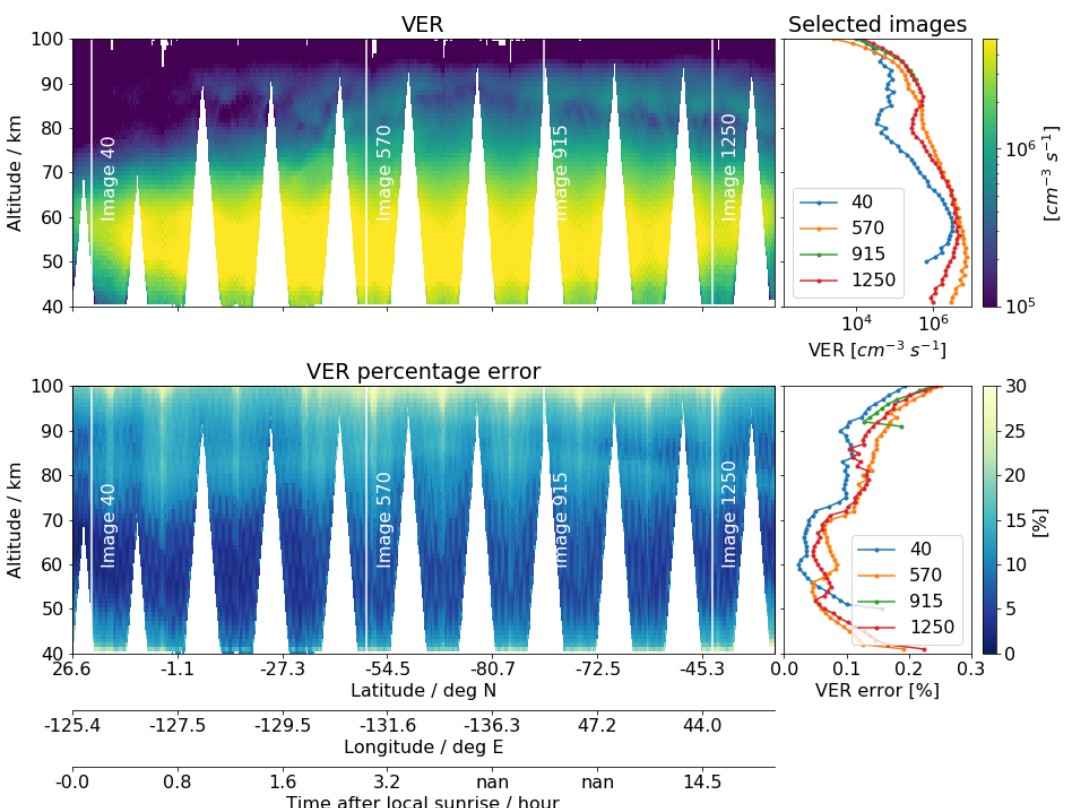

**Figure 4.** An example of the retrieved volume emission rate of $O_2(a^1\Delta_g)$ from IRI during daytime for one orbit collected on 2008-1-2, from 15:14:24 to 16:2:31 (orbit number 37400) and its random error relative to the individual a priori profile (upper and lower panels, respectively). The panels on the left are 2D-colour plots, as a function of geographical location and altitude. Time after local sunrise is also shown in the horizontal axis, where 'nan' indicates a location near the summer pole where the sunrise is absent. The panels on the right are the vertical profiles of the four selected images (along the four white lines on the left panels). All results shown here have a measurement response greater than 0.8.

layers can clearly be seen, separated by a minimum around 80 km and mixed with some finer structures between the layers. The relative random error is generally under 25% above 70 km and under 10% below 70 km. Figure 4 also shows data gaps in a 'zig-zag' pattern, which are due to the nodding motion of the Odin satellite to facilitate the limb scanning process of the other instruments onboard (OS and SMR). Specifically, in this particular orbit, the satellite was in the so-called 'mesospheric scan mode' which results in data gaps up to 95 km. Hence, not all profiles can reach as low as 40 km, as shown on the right panel of Fig. 4 (e.g. image 915). The first 3-4 nods in Fig. 4 correspond, however, to the 'normal scan mode' resulting in fewer data gaps.

## 2.3 Retrieval of ozone

Measurements of oxygen IRA band are often used as proxies to estimate daytime ozone concentration because the production of $O_2(a^1\Delta_g)$ is closely linked to the available ozone during the daytime. Measurements such as those from SME, SABER and SCIAMACHY have been used to estimate the ozone concentration using a chemical kinetic model and assuming photochemical equilibrium in a similar fashion (e.g., Thomas et al., 1984; Mlynczak et al., 2007; Zarboo et al., 2018).

Estimation of ozone concentrations from airglow observation highly relies on the assumption of photochemical equilibrium as well as an accurate chemical kinetic model that relates the volume emission rate to the ozone number density. One limitation of using the equilibrium assumption is the time delay in response to any change in ozone, due to the long lifetime of the airglow species. Thus such an approach will lead to an under- or over-estimation of ozone concentration if the equilibrium state of the airglow species is not yet reached at a given time and location. Odin takes measurements in a 6h-18h polar orbit, thus a considerable portion of the daytime orbit is close to the local sunrise and sunset, especially in the equatorial region. The closer to the sunrise the further the $O_2(a^1\Delta_g)$ is from the equilibrium state, because of the dominant source of the emission being the solar photolysis. As for the local sunset, the problem of equilibrium assumption arises mainly after the photolysis process has stopped, thus beyond the data range used for this study. In this paper, we apply a special treatment on the ozone retrieval near the local sunrise where the photochemical steady state assumption can not be considered as valid. This will be described in Sect. 2.3.3.

In addition, reaction processes, Einstein coefficients, reaction rates, photolysis rates, reaction efficiency, solar irradiance, and such must be described as correctly as possible in the chemical model. In the early 1980s, R. J. Thomas et al. (1983) developed a simple photochemical model which only included ozone photolysis in the Hartley band and solar excitation of $O_2$ in the atmospheric band. This model was applied to SME $O_2(^1\Delta)$ measurements to derive ozone. After that, Mlynczak et al. (1993) showed that the photolysis of $O_2$ in the Schumann-Runge continuum and Lyman alpha spectral region make significant contributions to the $O_2(a^1\Delta_g)$ production through $O(^1D)$ production at higher altitudes. They concluded that the previous model led to an over-estimation of the ozone concentration from $O_2(a^1\Delta_g)$ observations. Accompanying the launch of SABER, in 2001, this model was further updated, the radiative lifetime of $O_2(a^1\Delta_g)$ was revised and other minor modifications were made, in order to derive ozone profiles from SABER $O_2(a^1\Delta_g)$ measurements (Mlynczak et al., 2007). Yankovsky and Manuilova (2006) have concluded that supplementing the vibrational states in the comprehensive photochemical model helps to get a better agreement between the ozone profiles retrieved from 1.27 μm and 762 nm emissions, based on a numerical experiment on

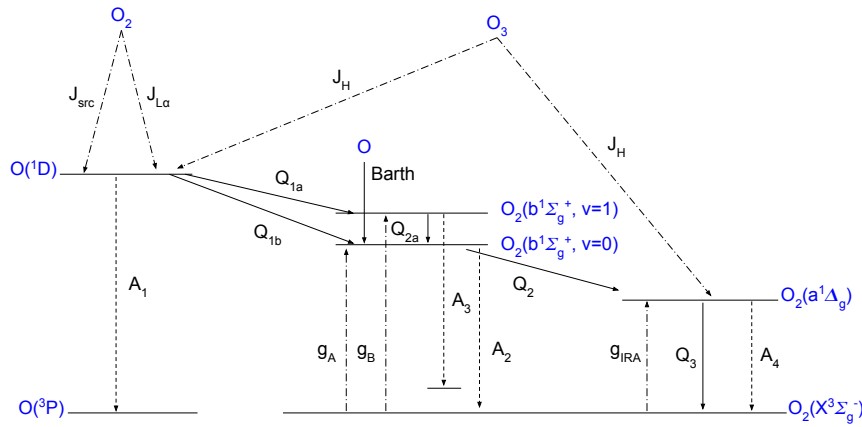

**Figure 5.** Scheme of kinetics of excited levels of atomic oxygen and molecular oxygen that are used in our model. Detailed reactions are described in Table A1

a few METEORS rocket profiles (Mlynczak et al., 2001). Their model includes 3 vibrational levels of $O_2(b^1\Sigma_g^+, v=0,1,2)$, 6 of $O_2(a^1\Delta_g, v=0-5)$ and 35 of $O_2(^3\Sigma, v=0-34)$. Yankovsky et al. (2016) have used the same model to simulate how various oxygen airglows perform as proxies for atomic oxygen and ozone.

### 2.3.1 The kinetic model

In this paper, we use a kinetic model with the inclusion of two vibrational levels of $O_2(b^1\Sigma_g^+, v=0,1)$, the Barth-type chemical
mechanism (McDade et al., 1986), as well as the solar resonance absorption in the oxygen IRA band itself, which can be described as a model whose the complexity is in between the one used by Mlynczak et al. (1993) and the one used by Yankovsky et al. (2016). Figure 5 illustrates the kinetic scheme of our model. Neglecting most of the vibrational sub-levels of each electronic state should not greatly affect the accuracy of the retrieved ozone. This can be considered as a reasonable assumption, because the population of the electronical-vibrational excited states is mostly dominated by the lowest vibrational state in each
electronic level, and these sub-levels are eventually quenched to the lowest vibrational levels as shown by Yankovsky and Manuilova (2006). The processes that we have considered in our kinetic model are listed briefly below, while detailed reactions, as well as the corresponding rate coefficients and quantum yields or efficiencies, can be found in Table A1.

- $J_H$: photodissociation of ozone in the Hartley band ($\lambda < 310\,\text{nm}$) produces the electronically excited state atomic oxygen $O(^1D)$ and molecular oxygen $O_2(a^1\Delta_g)$;

- $J_{SCR}, J_{L\alpha}$: photodissociation of ground state molecular oxygen in both the Schumann-Runge continuum ($130 \leq \lambda \leq 175\,\text{nm}$) and at Lyman $\alpha$ ($\lambda = 121.6\,\text{nm}$) produces ground state $O(^3P)$ and excited atomic oxygen $O(^1D)$;

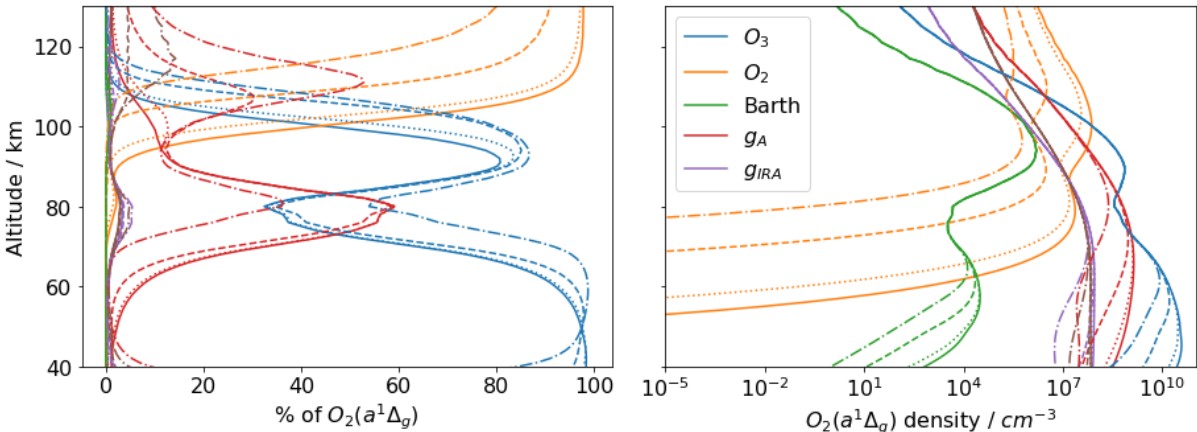

**Figure 6.** Left panel: relative contributions of 6 different sources to the production rate of $O_2(a^1\Delta_g)$ at 4 solar zenith angles, namely $30°$ (solid lines), $60°$ (dotted lines), $85°$ (dashed lines) and $89.9°$ (dot-dashed lines). Right panel: same as left, but in absolute concentration of $O_2(a^1\Delta_g)$.

- $g_A, g_B, g_{IRA}$: resonance absorption of ground state molecular oxygen at A-band (762 nm), B-band (688 nm) and oxygen IRA band (1270 nm), respectively;

- $Q_{1a}$: transfer of energy by quenching $O(^1D)$ with $O_2(^3\Sigma)$ partly forms ground state atomic oxygen $O(^3P)$ and an excited level of molecular oxygen $O_2(b^1\Sigma_g^+, v=1)$;

- $Q_{1b}$: same as $Q_{1a}$ but to form $O_2(b^1\Sigma_g^+, v=0)$;

- Barth: recombination between two oxygen atoms that through energy transfer processes produce $O_2(b^1\Sigma_g^+)$, known as the Barth-type mechanism;

- $Q_{2a}$: quenching of $O_2(b^1\Sigma_g^+, v=1)$ to the lower vibrational level $O_2(b^1\Sigma_g^+, v=0)$.

- $Q_{2b}$: quenching of $O_2(b^1\Sigma_g^+, v=0)$ to the lower electronically excited state of molecular oxygen $O_2(a^1\Delta_g)$;

- $Q_3$: quenching of $O_2(a^1\Delta_g)$ to the ground state $O_2(^3\Sigma)$;

- $A_1, A_2, A_3, A_4$: the inverse of photochemical lifetime of $O(^1D)$, $O_2(b^1\Sigma_g^+, v=0)$, $O_2(b^1\Sigma_g^+, v=1)$ and $O_2(a^1\Delta_g)$, respectively, when they eventually release their energy as a photon and transfer back to the ground electronic state.

Figure 6 shows the contributions from different production sources to $O_2(a^1\Delta_g)$ in the altitude range 60 to 150 km, both in percentage and absolute concentrations. The simulation is based on a single ozone profile taken from CMAM, a background density and a temperature profile taken from the MSIS climatology at different solar zenith angles. Percentage-wise, photodissociation of molecular oxygen in both Schumann-Runge continuum and at Lyman $\alpha$ dominate above 100 km, which is consistent with Mlynczak et al. (1993). Below 100 km $O_2(a^1\Delta_g)$ is mainly produced by photodissociation of ozone in the

Hartley band as well as by resonance absorption in the A-band. Resonance absorption in the oxygen B-band and oxygen IRA
band contribute as much as 6% at around 80 km or even higher at around 115 km. Moreover, $O_2$ photodissociation in the altitude range of 60-150 km is highly sensitive to the solar zenith angle while the other photochemical sources are only sensitive below 70 km, except for the solar excitation of $O_2$ being below 100 km. The Barth-type mechanism contributes very little and mainly between 90-105 km. However, the Barth-type mechanism is the only source during the absence of sunlight since all the other sources involve photochemical reactions, which explains why the nightglow is much weaker than the dayglow (not
shown in this paper).

Without simultaneous measurements of $O_2(b^1\Sigma_g^+)$, a reasonable assumption on the efficiency of $O(^1D)$ quenched by ground state $O_2$ to $O_2(b^1\Sigma_g^+)$ is needed. We assume that 20% are quenched to $O_2(^1\Sigma), v=0$ and that the rest are quenched to $O_2(^1\Sigma), v=1$, as indicated by Yankovsky et al. (2016). All $O_2(^1\Sigma), v=1$ are assumed to be quenched by $O_2$ and $N_2$ to $O_2(^1\Sigma), v=0$. Uncertainties in other reaction rate coefficients and their sensitivity to the retrieved ozone concentration are
further discussed in Yankovsky et al. (2016).

By assuming photochemical equilibrium for $O(^1D)$, $O_2(b^1\Sigma_g^+)$ and $O_2(a^1\Delta_g)$, one may establish a system of equations to solve for the ozone concentrations from the measured oxygen IRA band volume emission rate. However, it is not straight forward to simply invert the system of equations as the model is non-linear. For example, calculating the solar photolysis rate in the Hartley band and calculating the atomic oxygen density for the Barth-type mechanism depend on how much ozone is
present.

### 2.3.2 The inversion method

We choose to use the Levenberg-Marquardt method to retrieve the ozone number density iteratively (Rodgers, 2000). The ozone number density $\mathbf{x}$ at each iteration $n + 1$ is derived using the formula

$$\mathbf{x}_{n+1} = \mathbf{x}_n + [(1+\gamma)\mathbf{S_a}^{-1} + \mathbf{K}_n^T \mathbf{S_e}^{-1} \mathbf{K}_n]^{-1} \mathbf{K}_n^T \mathbf{S_e}^{-1} [\mathbf{y} - \mathbf{F}(\mathbf{x}_n)] - \mathbf{S_a}^{-1} [\mathbf{x}_n - \mathbf{x_a}], \tag{15}$$

where $\gamma$ is the damping parameter and $\mathbf{y}$ is the previously estimated volume emission rate profile (i.e. $\hat{\mathbf{x}}$ in Sect. 2.2) with a measurement response larger than 0.8.

In our implementation, all negative volume emission rates are treated as invalid and replaced by an interpolated value. $\mathbf{F}(\mathbf{x}_n)$ is the volume emission rate evaluated by the photochemical model and $\mathbf{K}_n$ is the numerically calculated Jacobian at the n-th iteration based on $\mathbf{F}(\mathbf{x}_n)$. All negative ozone number densities are forced to be $10^{-8}\,\mathrm{cm}^{-3}$ in $\mathbf{F}(\mathbf{x})$. $\mathbf{S_e}$ is a diagonal matrix
which refers to the result of previous retrieval step, being the retrieval noise of the volume emission rate (i.e. $\mathbf{S_m}$ in Eq. 14). However, this measurement uncertainty matrix is further modified to address the issue of the validity of the photochemical equilibrium assumption at a given time and location. The details will be discussed in Sect. 2.3.3. All off-diagonal elements in $\mathbf{S_m}$ are removed because the inversion of the full matrix often leads to numerical instability. We use the ozone profiles taken from the CMAM climatology as $\mathbf{x_a}$. $\mathbf{S_a}$ follows the same formula as in the retrieval of volume emission rate (see Eq. 2.2),
also with $\sigma_a = 0.75\mathbf{x_a}$. As the Levenberg-Marquardt method is an iterative procedure to solve non-linear problems, it requires an initial guess. We use $\mathbf{x_a}$ for this. The background air density and temperature are taken from the MSIS climatology. The

volume mixing ratio of $O_2$, $N_2$ and $CO_2$ are assumed to be 21%, 78% and 405 ppm, respectively, at all altitudes. For the number density of O, we assume photochemical steady state with ozone.

When iteration has converged, $\gamma$ is mostly sufficiently small ($\gamma \ll 1$) such that the retrieval can be approximated by using a Gauss-Newton method at the final iteration. Thus the relevant equations for the error analysis are essentially the same as Eq. 9 and Eq. 14 described in Sect. 2.2. Similarly, AVK and MR can be assessed using the Jacobian matrix at the final iteration. Finally, the normalised cost of the retrieval is evaluated as

$$\chi_n^2 = [(\mathbf{x}_n - \mathbf{x_a})^T \mathbf{S_a}^{-1}(\mathbf{x}_n - \mathbf{x_a}) + (\mathbf{y} - \mathbf{F}(\mathbf{x}_n))^T \mathbf{S_e}^{-1}(\mathbf{y} - \mathbf{F}(\mathbf{x}_n))]/m \tag{16}$$

where $m$ is the number of elements in $\mathbf{y}$ vector (here is the same as in $\mathbf{x}$ vector, i.e. number of atmospheric layers).

### 2.3.3 The photochemical equilibrium assumption

The inversion process described above highly relies on the assumption of photochemical steady state. As previously mentioned, if the $O_2(a^1\Delta_g)$ has not yet reached its equilibrium state since the start of its production, such an assumption will lead to an under-estimation of the derived ozone near the local sunrise. The reason for this under-estimation is that since the $O_2$ contribution to the production of $O_2(a^1\Delta_g)$ is fixed, the low measured intensity of the 1.27 µm volume emission rate ends up being compensated as low or even negative values of ozone in the inversion process.

A considerable portion of the IRI measurements do occur close to the day-night terminator and are therefore affected by this problem. In this section, we describe an extra step of the retrieval process intended to address this divergence from equilibrium when necessary. The approach employed allows us to deal with the "turn on" of the $O_2(a^1\Delta_g)$ production at sunrise. However, it will not compensate for the time delay associated with changes in ozone throughout the day where we will always have an extra source of uncertainty.

Figure 7 attempts to illustrate a simple estimation of the ozone number density assuming photochemical equilibrium for all IRI measurements in one orbit. As the orbit proceeds (from left to right in Fig. 7), the effects of under-estimation of ozone can be seen where the measurements are made closer and closer to the local sunrise.

The change of the number density $[O_2(a^1\Delta_g)]$ can be described by a dynamical equation

$$\frac{d[O_2(a^1\Delta_g)]}{dt} = P - L[O_2(a^1\Delta_g)] \tag{17}$$

$$= P - \frac{[O_2(a^1\Delta_g)]}{\tau}, \tag{18}$$

where $t$ is the time since the production has started (i.e. time after the local sunrise), $P$ represents the production terms and $L$ the loss rate of $O_2(a^1\Delta_g)$. The loss rate which is also the inverse of lifetime $\tau$, consists of two components, the radiative relaxation and the collisional quenching ($A_4$ and $Q_3$ in Fig. 5). Assuming $P$ and $L$ are independent of time and $[O_2(a^1\Delta_g)]$ starts from zero when $t = 0$, one can provide a solution to the ordinary differential equation as

$$[O_2(a^1\Delta_g)] = [O_2(a^1\Delta_g)]_{equi}(1 - \exp(-t/\tau)), \tag{19}$$

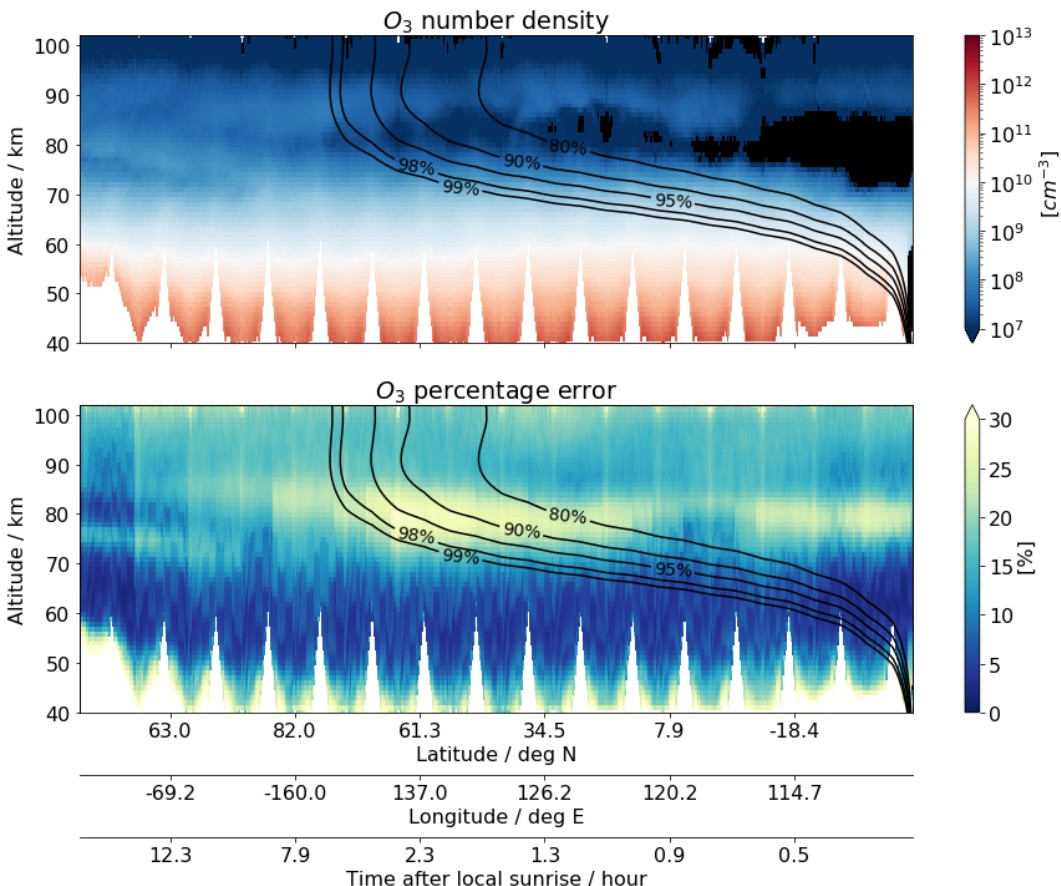

**Figure 7.** A simple estimation of ozone number density (upper) and its random error relative to the individual a priori profiles (lower), assuming all collected measurements of $O_2(a^1\Delta_g)$ are in equilibrium state, for one orbit collected on 2008-03-30, from 22:21:09 to 23:09:13 (orbit number 38720). Superimposed the contour lines of equilibrium index (see text) corresponding to 80%, 90%, 95%, 98% and 99% of the equilibrium level. Negative values are indicated as black colour. All results shown here have a measurement response greater than 0.8.

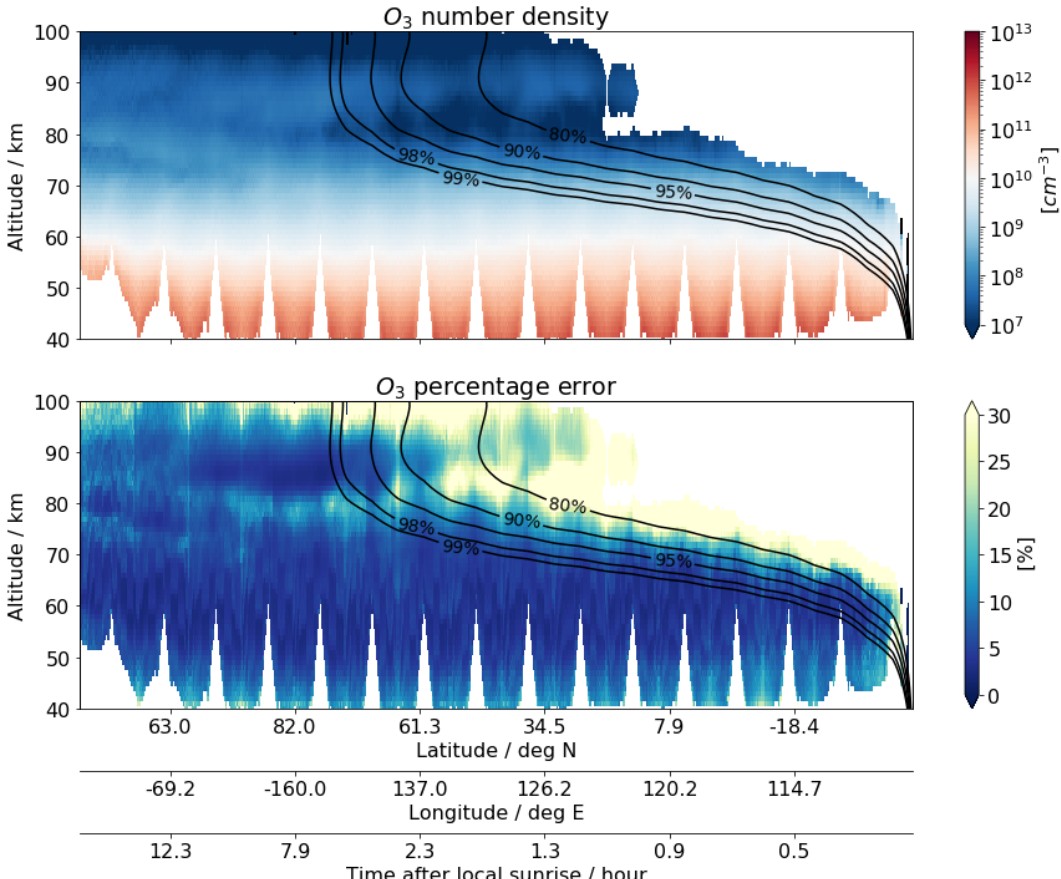

**Figure 8.** Same as Fig. 7, but showing the final estimation with the adjustment of the equilibrium index.

where $[O_2(a^1\Delta_g)]_{equi} = P/L = \tau P$. Since $A_4$ is a fixed coefficient and $Q_3$ is mainly based on the background density for $O_2(a^1\Delta_g)$ kinetics, the lifetime $\tau$ can easily be calculated as a function of altitude. $L$ is dominated by $A_4$ above 75 km while below is dominated by $Q_3$. As the ratio $t/\tau$ takes the values of 1.6, 2.3, 3, 4 and 4.6, $[O_2(a^1\Delta_g)]$ has reached 80%, 90%, 95%, 98% and 99% of the equilibrium level, which are indicated by the contour lines superimposed on Fig. 7. Thus, we can use $(1 - \exp(-t/\tau))$ as an equilibrium index to indicate how far the given time and location is from the equilibrium state.

To address the validity of the equilibrium assumption in the ozone retrieval, the uncertainty of the measurement vector $\mathbf{S_e}$ is further modified as

$$\mathbf{S_e}^{modified} = \mathbf{S_e}/(1 - \exp(-t/\tau))^8, \tag{20}$$

where the equilibrium index is raised to the power of 8 in order to force a sufficiently low measurement response in the relevant time and altitude ranges so that the affected data can be filtered out.

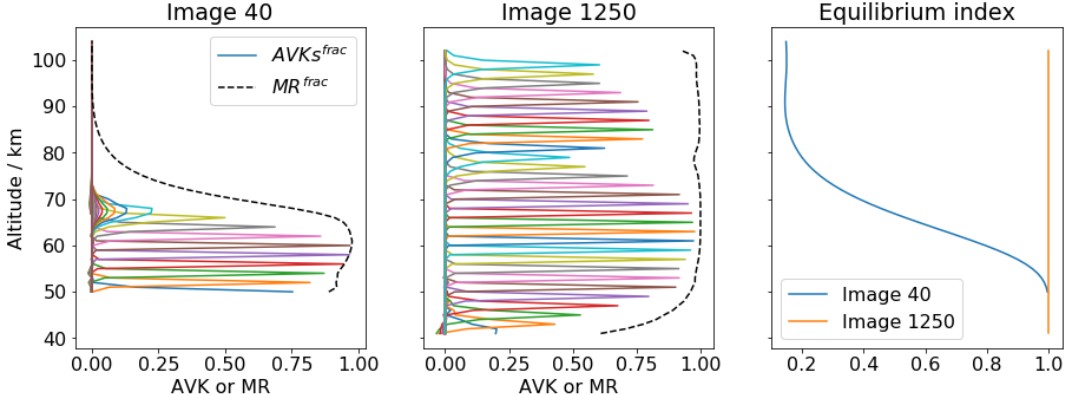

**Figure 9.** Every second row of the fractional AVK (solid lines) and MR (black dashed line) of the ozone retrieval for two example images (the first two panels) taken from the orbit number 37400 (see Fig. 4 for reference). The corresponding equilibrium indices of these two images are shown in the third panel.

As a result of such a modification, the inversion will avoid updating the a priori profile giving a low measurement response in the region where the equilibrium assumption is far from valid, while $S_e$ is barely modified where the equilibrium index is close to 100%. As shown in Fig. 8, the upper right region of the plots, where the equilibrium index is significantly lower than

100%, is blanked out due to the low measurement response, while the lower left part of the plots show no difference to the ones shown in Fig. 7. In addition, such an adjustment on $S_e$ amplifies the represented random error where the equilibrium index is relatively lower, indicating that this part of the data set should be handled with care.

Two example images of the fractional AVK and MR are shown in Fig. 9, which correspond to the images 40 and 1250 in orbit number 37400 (see Fig. 4). The first example image clearly shows that the measurement response is effectively dampened

by $S_e^{modified}$ with a low equilibrium index above $65\,km$, while the second example image keeps a high measurement response at almost all altitudes. The full width at half maximum of the AVKs indicates that the vertical resolution of the ozone profiles is about $1$-$2\,km$ where the data points are considered to be valid. Note that the AVKs above $90\,km$ may not necessarily represent the 'true' values as the retrieval resolution of the volume emission rate is not properly taken into account in the ozone retrieval. Thereafter we present our results for all IRI ozone data points that have a fractional MR greater than 0.8, $\chi^2$ smaller than 10

and equilibrium index corresponding to more than 95% of the equilibrium value. Also, the lowest $10\,km$ grids in the retrieval are filtered out to avoid biases due to the possible edge effect. After all these criteria are used for filtering, the IRI ozone data availability at $80$ $km$ over one year (every 20th orbit) is presented in Fig. 10. A significantly high number of profiles is located in the summer polar region because of the 6-18h Odin orbit. Nearly no data is available at the tropics, at the altitude of $80$ km, due to the fact that the measurements were made too close to sunrise.

Overall, the resulting IRI mesospheric ozone product has a precision of around 5-20% based on the retrieval noise estimate (see bottom panel of Fig. 8), with relatively larger values above $80\,km$ and below $50\,km$. However, the systematic error in the estimated ozone product is as large as 50% as seen in the comparisons with other ozone data sets (see Sect. 3.4). A

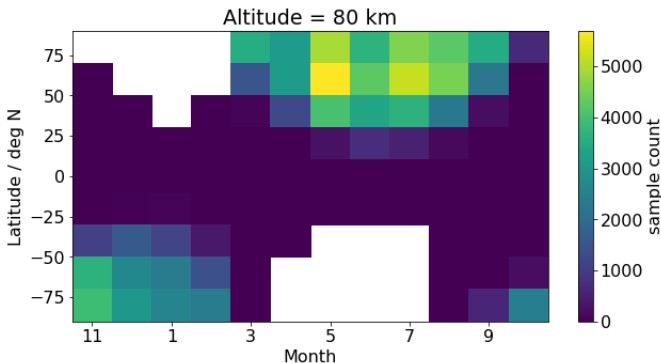

**Figure 10.** IRI ozone data availability at 80 km for a one-year sample data set (every 20th orbit), after validity criteria have been applied (see text). The number is expected to increase by 20 times when all the orbits of one year are processed.

**Table 1.** A summary of the possible sources of systematic error in the IRI ozone and their relative errors.

| Error sources | Estimated errors |
|---|---|
| Kinetic parameters in the photochemical model[a] | <20% below 90 km, 20-100% above 90 km |
| Absolute calibration[b] | <20% |
| Time delay of $O_2(a^1\Delta_g)$ in response to changes in $O_3$ | 10-20%[c] |
| Instrumental pointing | 8-15%[d] |
| Absorption correction factor | <1% |
| Temperature dependent filter overlapping | <1% |

a: see Yankovsky et al. (2016) as well as Mlynczak and Olander (1995) for a comprehensive analysis on the sensitivity of each parameter

b: due to lack of in-flight calibration

c: see Zhu et al. (2007)

d: estimated from 250 - 500 m bias in the pointing uncertainty

detailed investigation of the reasons for these differences may be carried out in a future study when the 20-year-dataset has been processed. Moreover, the precise quantification of these systematic error sources requires detailed modelling studies and thus the foci of this paper remain on the retrieval technique on deriving daytime ozone from the 1.27 μm emission. Here, we summarise the possible sources to the systematic error and estimate their relative values to our best knowledge for the potential future data users (see Table 1).

## 3   Ozone comparisons

To illustrate the performance of the technique described in Sect. 2, daytime ozone profiles have been derived for a test sample of 5 % (every 20th orbit) of all the limb measurements collected by IRI from November 2007 to October 2008. In order to show the consistency of the results, these IRI ozone profiles are compared with independent data sets derived from OS, SMR

**Table 2.** Main characteristics of the ozone data sets under consideration

| Instrument (satellite) | Version | Precision | Vertical resolution | Vertical coverage | measured quantity |
|:---:|:---:|:---:|:---:|:---:|:---:|
| IRI (Odin) | V1-0 | 5-20 % | 1-2 km | 50-100 km | ND[a] |
| OS (Odin) | V5-10 | <5 % | 2-4 km | 10-60 km | ND |
| SMR (Odin) | V3.0.0 | ~1 ppmv | 3.5 km | 12-95 km | VMR[b] |
| MIPAS (Envisat) | V5R | 5-10 % | 4-8 km | 5-100 km | VMR |

a: Number density

b: Volume mixing ratio

and MIPAS. We would like to emphasise that a comprehensive validation study is not the primary intention of this study. This will be a valuable future study, after the whole 20-year IRI data set has been processed. We choose to use mainly number density for the comparisons, as it is the natural unit of the IRI and OS ozone profiles. As pointed out by Smith et al. (2013), the
differences in background densities to derive ozone VMR introduce additional uncertainty between instruments. As such, we would like to avoid using external data as much as possible. Moreover, the measurements of ozone by SMR at higher altitudes are mostly based on Doppler broadened lines, thus the natural unit is closer to number density rather than VMR. Therefore, the visualisation of the profile comparisons in this section is shown in ozone number density, with only one exception in the last panel of Fig. 11.
The data sets under consideration are briefly described in Sect. 3.1. In Sect. 3.2, we compare coincident observations made by OS, IRI and SMR, all from one example orbit. Since they are on board the same spacecraft, numerous co-incident profiles can be found. Yet, the measurement principles of these three instruments are intrinsically different. In Sect. 3.3, we will focus on the annual cycle of daytime ozone vertical structures in the MLT region. Lastly, the comparison of the zonally averaged daytime ozone profiles from the four aforementioned instruments is discussed in Sect. 3.4. Our goal is to illustrate the consistency of
IRI ozone profiles with the other ozone products and, to a lesser extent, to interpret the differences between them.

### 3.1 Other ozone data sets

Although this is not a complete validation study, independent ozone data sets are used to compare with the new ozone product. The main characteristics of these data sets are given in Table 2. We have selected measurements made between November 2007 and October 2008 by each of the instruments under consideration. A brief description of the measurement principles and data
screening methods is provided in this section. More detailed information can be found in the cited publications.

### 3.1.1 OS

In previous publications, the term OSIRIS ozone product usually refers to the product obtained from the optical grating spectrograph (e.g., McLinden et al., 2007; Bourassa et al., 2018). In Smith et al. (2013), OSIRIS ozone refers to the product derived from the A-band airglow emission by Sheese (2009). In this paper, we will use OS ozone to refer to the product derived from
the measured limb scattered sunlight in the Chappuis and Hartley-Huggins bands. These ozone profiles are retrieved from limb

radiance pairs and triplets using the multiplicative algebraic reconstruction technique (MART) (Degenstein et al., 2009). This data set is one of the OSIRIS operational products within the ESA Climate Change Initiative (CCI) programme [2]. Invalid values have already been screened out by the instrument team (Sofieva et al., 2013).

### 3.1.2 SMR

The sub-millimetre radiometer on-board the Odin satellite measures spectra at different altitudes during the limb scans. In particular, it measures the ozone thermal emission line at 545 GHz (this is the so-called frequency mode 2 in the SMR nomenclature). As described in Eriksson (2017), vertical profiles of ozone are retrieved based on the optimal estimation method (OEM) by inverting the radiative transfer equation for a non-scattering atmosphere. This SMR ozone data set has recently been reprocessed. In this study, we use the new ozone main product of SMR, whose quality was assessed in Murtagh et al. (2018).

This product is in much better agreement with other instruments, compared to the previous version. All data points that have a measurement response lower than 0.8 are considered as invalid values. Ozone volume mixing ratio (VMR) is provided and the ozone number density is determined by multiplying it with the background number density provided in the data set. This background number density comes essentially from ERA-interim up to 60 km, NRLMSISE-00 from 70 km, and a spline interpolation between 60 and 70 km. All nighttime measurements (i.e. with the labelled solar zenith angle larger than $90°$) are

screened out for analysis as we only look at the daytime ozone distribution in this paper, except for in Fig. 11 for demonstration purpose.

### 3.1.3 MIPAS

The Michelson Interferometer for Passive Atmospheric Sounding measures the thermal emission band of ozone at $9.6\,\mu\text{m}$. We chose to use the middle atmospheric mode in our analysis. This data set has been processed by KIT-IMK and IAA-CSIC and

documented in Van der A et al. (2017); López-Puertas et al. (2018). As for SMR, all nighttime measurements are excluded from further analysis. Following the MIPAS level 2 screening recommendations, all data points that are flagged by 'visibility = 0' or have an averaging kernel diagonal element of less than 0.03 are not considered. Ozone concentration is given in VMR. We use the temperature and pressure measured by MIPAS (García-Comas et al., 2014) to calculate the ozone number density.

### 3.2 Comparison of coincident profiles

As mentioned in the earlier sections, the Odin satellite collects ozone profiles from three independent instruments. SMR measures thermal emission of an excited state of ozone in the microwave region at 545 GHz, OS measures scattered solar light in the Hartley-Huggins and Chapuis bands and IRI measures the oxygen airglow emission at oxygen IRA band. Due to the underlying measurement principles of these data sets, the altitude ranges and parts of the orbit during which data is available vary. This is depicted in Fig. 11. For this particular example orbit, SMR measures ozone from 15 to a maximum 75 km altitude

both daytime and nighttime throughout this particular orbit, while OS measures ozone only up to 55 km and during only half

---

[2]ESA CCI programme: http://cci.esa.int/ozone

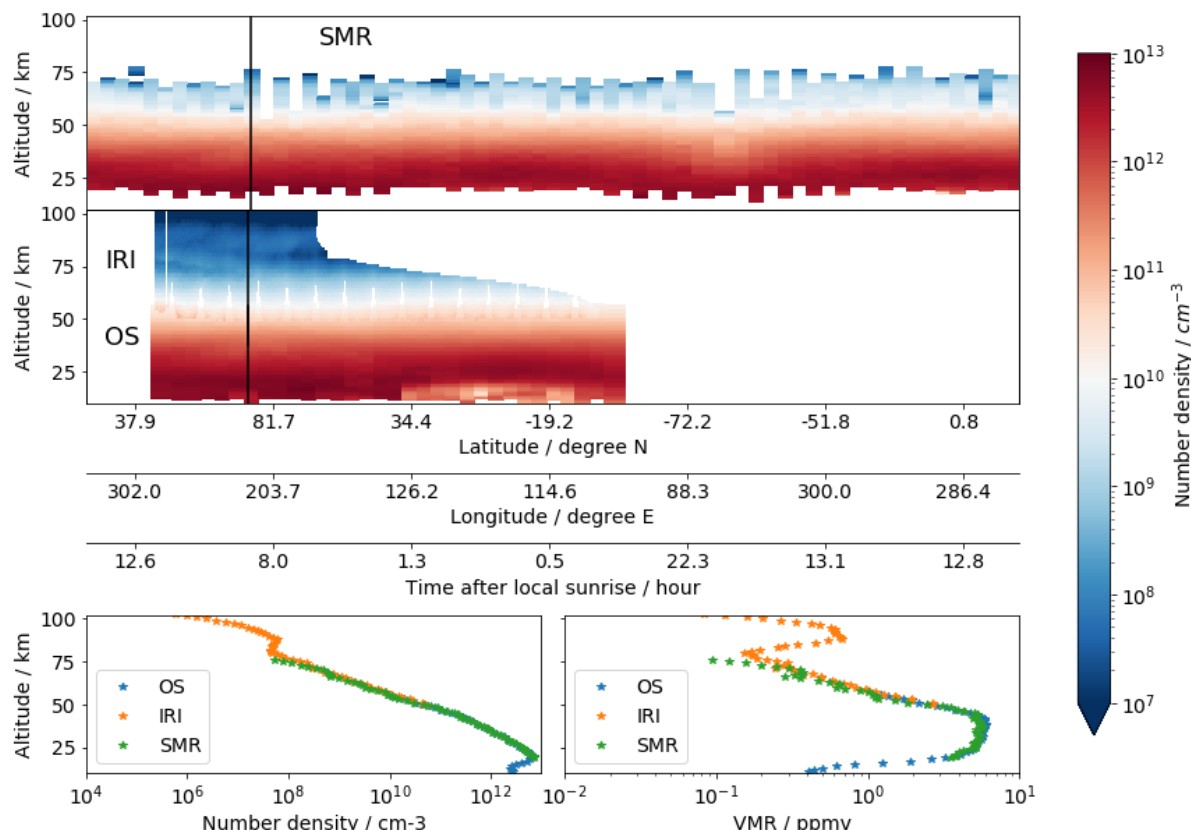

**Figure 11.** Upper panel: 2D-colour plot of SMR ozone number density profiles for one orbit as a function of altitude and geographical location, collected on 2008-3-30 from 22:14:56 to 23:51:03 (Odin orbit number 38720). Middle panel: 2D-colour plot of IRI (above 50 km) and OS (below 60 km) daytime ozone profiles corresponding to the same orbit as SMR. Bottom left: ozone number density profiles from the scan/image of OS, IRI and SMR indicated by the black vertical line in the panels above. Bottom right: same as the bottom left panel but showing the volume mixing ratio in ppmv.

of the orbit, where scattered sunlight is available. IRI ozone covers the altitude range from 50 to 100 km as limited by the VER retrieval grid and measurement response (see sections 2.2 and 2.3). IRI ozone is also limited to the day part of the orbit since the oxygen airglow primarily relies on photochemical reactions, as discussed in Sect. 2.3.

It is worth mentioning that Fig. 11 also illustrates the particularly high sampling rate of IRI, over 70 times higher than SMR and OS thanks to the imaging sensor. The bottom left panel of Fig. 11 shows a single ozone density profile collected at the same time and location by all three instruments. The volume mixing ratio of IRI ozone is derived based on the background density included in the SMR product (a combination of ERA-interim and NRLMSISE-00, see Sect. 3.1.2) and is shown in the bottom right panel of Fig. 11. Both the primary and secondary ozone layers appear clearly in the plots. These two plots suggest that these coincident ozone profiles would merge smoothly with each other, even though they do not cover the same altitude

ranges. While this is a single profile comparison, our general conclusion is that this holds for the majority of the profiles that we have inspected. This result shows how consistent the ozone observations from these three instruments aboard Odin are with each other, despite the fact that they use intrinsically different measurement techniques, even though the agreement between IRI and SMR ozone relatively worsens above 65 km. However, if one meticulously studies the 2D-colour plot of the IRI ozone in the mid-panel of Fig. 11, some vertical stripes may appear following the scanning pattern. We think that this effect is a result

of the stray light correction process in the level 1 data (See Sect. 2.1.3).

### 3.3  Monthly mean time plots comparison

In this section, we show the monthly mean daytime ozone distribution in the MLT region as presented in an altitude-time plot for different latitude bands. We will look at the data sets in ozone number density from three instruments side-by-side, namely IRI, SMR and MIPAS.

Figure 12 shows the monthly mean ozone number density in the MLT region for six latitude bands: 70-90°, 50-70° and 30-50° in both hemispheres. The low latitude region is not shown due to the lack of IRI data above 80 km as expected from Fig. 10. It is recognisable that IRI ozone data set can reproduce the general seasonal pattern in the MLT, similarly to the other data sets. For all instruments at high-latitude, the top of the primary ozone layer extends to a higher altitude in the summer months, while being relatively stable when moving closer to the equator. This is associated with the large scale Brewer-Dobson

circulation of stratospheric ozone as discussed in Kyrölä et al. (2010). The secondary ozone maximum in the mesosphere at high-latitudes is located at roughly the same altitude of ca. 90 km among these ozone data sets. At mid-latitudes, the secondary ozone layer is slightly lower in altitude (ca. 85 km). This is consistently observed by all three instruments. A deep ozone trough between the main and the secondary layers is observed by MIPAS in the winter months in the high-latitude bands, while IRI and SMR lack of data in those regions because Odin was orbiting in the night part of the orbit. Overall, IRI agrees well with

MIPAS, and, to a lesser extent, with SMR due to the fact that very low (sometimes negative) values exist in the regions between the secondary and primary ozone layer in the SMR data. All instruments display a weaker secondary maximum in the lower latitude than in higher latitude regions, which is also shown by Smith et al. (2013) and may be explained by the tidal effects as mentioned in López-Puertas et al. (2018). We discuss the differences between these three instruments in more details in the next section.

### 3.4  Latitudinal distribution

Here, we focus on a selected month when IRI has a reasonably good latitudinal coverage to compare our newly derived data set with OS, SMR and MIPAS ozone in a more detailed manner, by looking at both a side-by-side comparison of the general global distribution and relative differences in different latitude bands. Similar conclusions can be drawn from the comparison of the other months of the year (not shown here).

Figure 13 depicts the daytime ozone distribution in number density, as observed in July 2008 by the four instruments under consideration, averaged in 10° latitude bins (upper panels). The 2D-histograms of the sample count in each latitude-altitude bin is also shown in the lower panels. The overall side-by-side comparison demonstrates that IRI data set is capable of representing

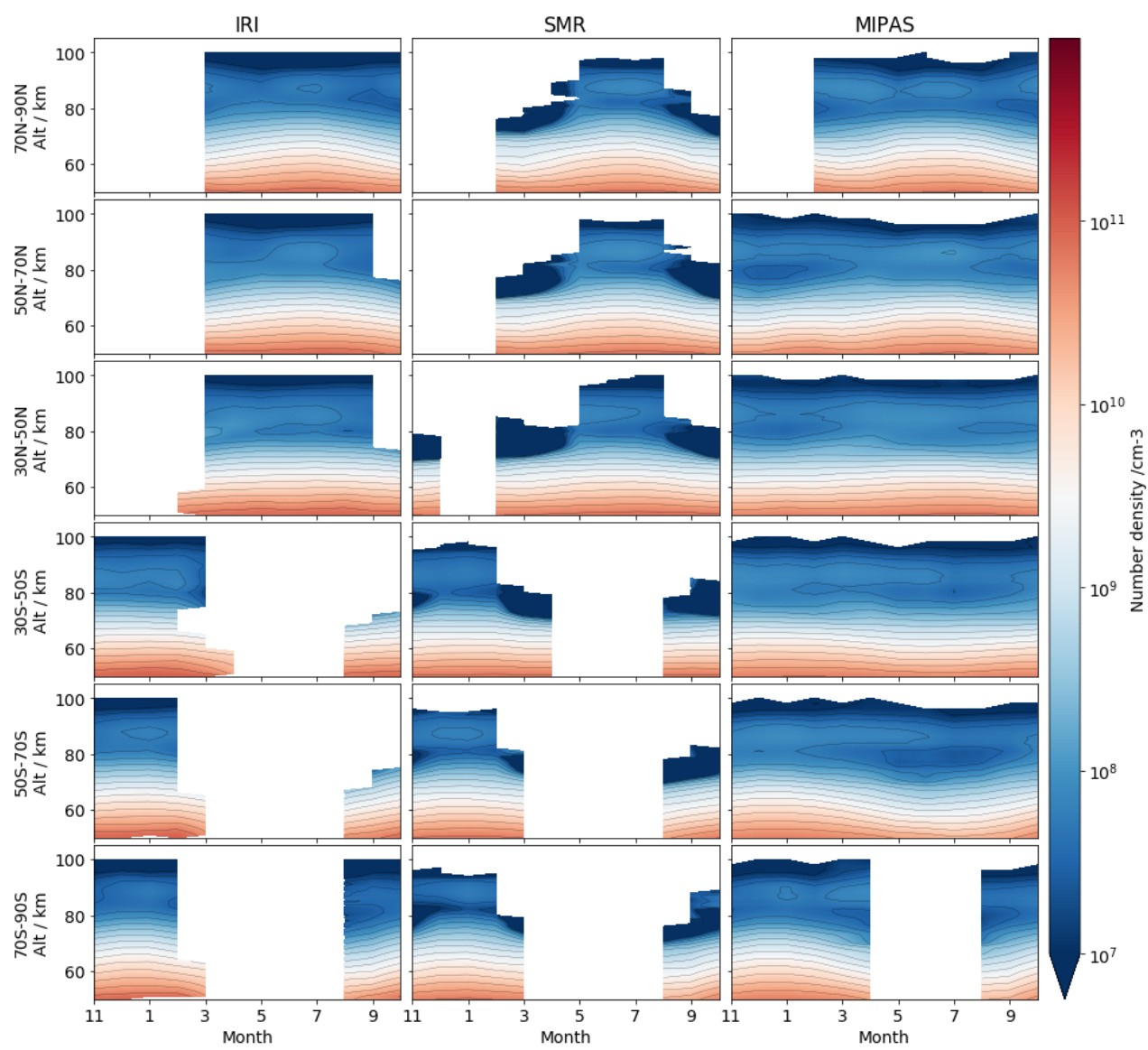

**Figure 12.** Monthly mean daytime ozone number density from November 2007 to October 2008, in six selected latitude bands (in rows). Columns from the left to right represent IRI, SMR and MIPAS, respectively.

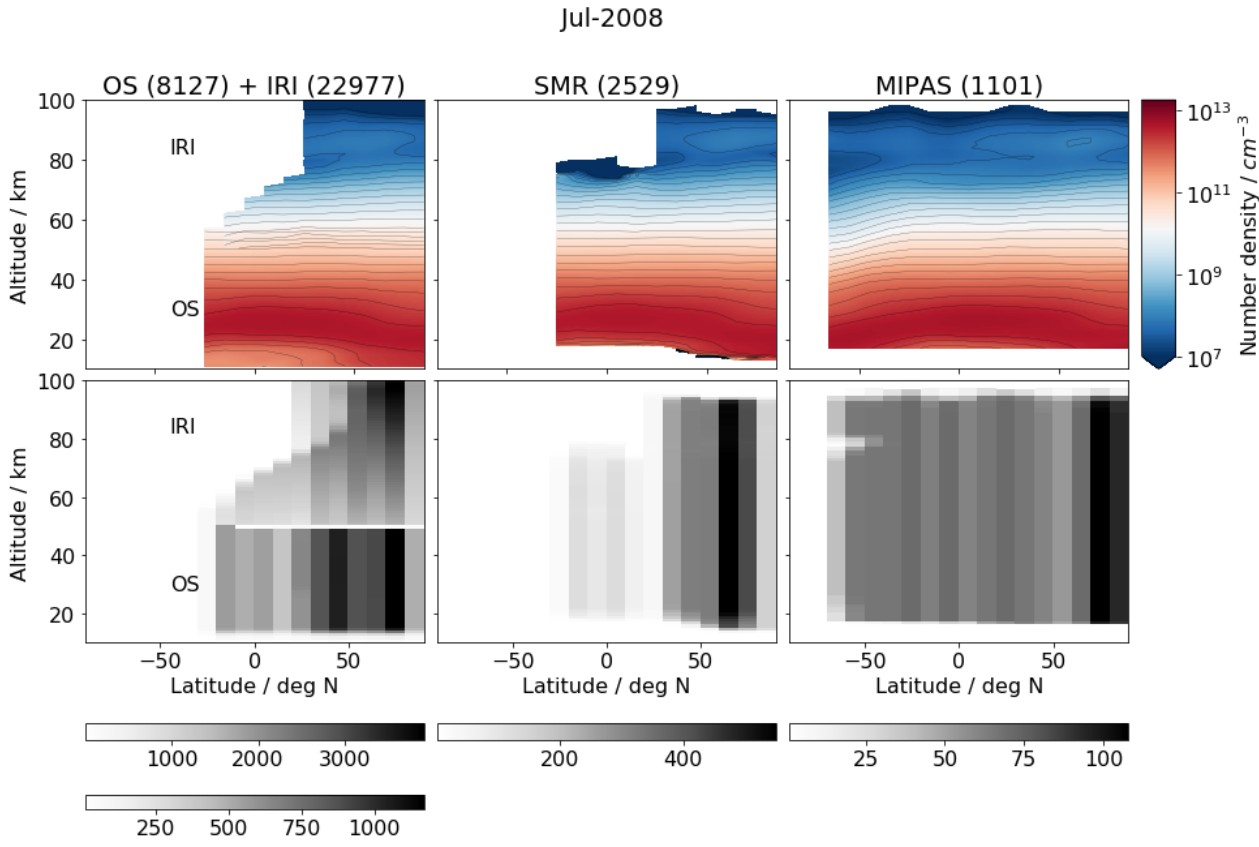

**Figure 13.** Zonally averaged daytime ozone number density observed in July 2008, with $10\,^{\circ}$ latitude bins. The first column contains data from two instruments, with OS below $60\,\mathrm{km}$ and IRI above $50\,\mathrm{km}$. The second and third columns correspond to SMR and MIPAS, respectively. The numbers in brackets indicated in the titles are the total number of profiles that are available within the month under consideration. Bottom panels correspond to 2D-histograms showing the number of samples that are accounted for the zonal average bins. Note that the grey scales of 2D-histograms are different in the bottom panels. The upper grey scale of the 2D-histogram on the left corresponds to IRI (>50 km) and the bottom one corresponds to OS (<50 km).

the general latitudinal distribution of ozone in the MLT, and complements well the already existing OS data set with the potential to merge with it. The secondary ozone layer around 90 km that peaks in the summer high latitude region can be observed in IRI, SMR and MIPAS, as expected from the seasonal trend shown in Fig. 12. SMR shows a region of very low or even negative values due to low measured signal around 80 km. Apart from the peak at 90 km at high latitudes in the summer hemisphere, another weaker peak can be observed in the winter hemisphere in the MIPAS data set. Unfortunately, this can not be observed by IRI and SMR due to the lack of daytime measurements in that region because of the Odin orbit.

Although the vertical coverage of the individual profiles may differ, IRI shows a great advantage with a significantly higher sampling rate than the other instruments, which implies that, when averaging large samples, the random error can be greatly reduced. Note that only every 20th orbit of IRI measurements has been processed in this study. Once all orbits are processed, the number of profiles are expected to be roughly 20 times higher. The 2D-histograms show that the sample size is significantly larger at high Northern latitudes than in the equatorial region for all instruments on board Odin, while MIPAS has a more evenly spread observation distribution, with still more data near the summer pole. Also, due to the equilibrium index filtering (see Sect. 2.3.3), IRI loses most of the data above 70 km in the tropics, since they were mostly measured very close to the local sunrise. As mentioned in Sofieva et al. (2014), insufficient or inhomogeneous sampling can result in inaccurate average estimates. However, a complete investigation of the sampling uncertainty is beyond the scope of this paper.

Figure 14 shows the relative differences between IRI and the other data sets, i.e. (IRI - Instrument)/IRI, for the ozone zonal mean profiles measured in July 2008 (other months show similar results). A general positive bias in IRI, with some exceptions at higher altitudes, can be seen in this figure. In the region below 70 km, IRI has a positive bias of up to 25% compared to all three instruments. Above 70 km, a positive bias of up to 50% is observed around 75 km compared to MIPAS, and similarly to SMR but observed around 80 km. However, between 80 to 90 km, negative differences of about 25% are observed compared to MIPAS, with an exception in the latitude band 20-30 deg N where the differences are bigger, up to -70%. Note that the biggest relative differences observed are at the lowest ozone concentration. In comparison with SMR, between 85 to 95 km, the differences vary from -25% to +10% (except for latitudes above 80 deg N, with differences up to -50%) depending on the latitude bands, with a larger negative bias to the north. Above 95 km, the relative differences are amplified due to the very low ozone densities and low sensitivity of the instruments, which is noted by the sensitivity analyses in Mlynczak and Olander (1995) and Yankovsky et al. (2016).

The uncertainties in the photochemical kinetic and spectroscopic rate coefficients, as well as the lack of in-flight absolute calibration, may be the main reasons for the general positive bias being observed, as listed in Table 1. In addition, as discussed in Zhu et al. (2007), the influence of transport e.g. tidal effects can be significant above 90 km as their amplitudes are large, which may lead to a significant error. Also, they point out that at around 80 km where $O_2(a^1\Delta_g)$ concentration is at its minimum and has a high vertical gradient, the transport term in the continuity equation become important, which is neglected in our Eq. 17. The abundance of the coincident measurements between IRI, SMR and OS provides a unique opportunity for a future investigation about the accuracy of the photochemical model. The absolute calibration error can be investigated after a more statistically significant amount of data will be processed (e.g. 19-year-data). López-Puertas et al. (2018) have reported that, in the summer months, MIPAS ozone has a negative bias of 20 to 80% between 60 to 85 km compared to SABER and

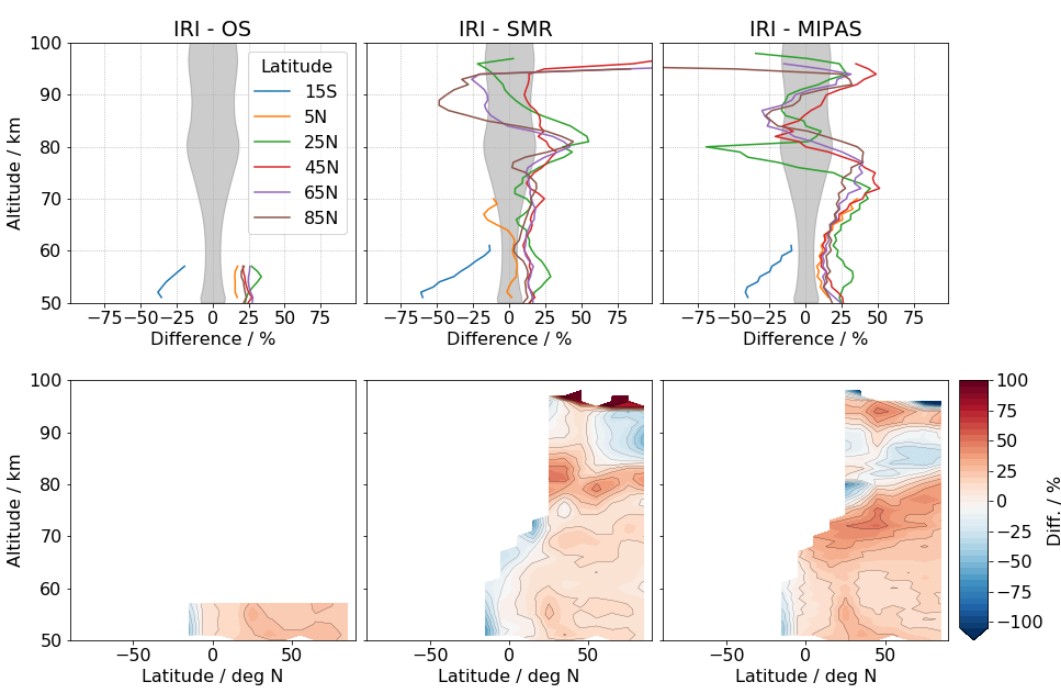

**Figure 14.** Relative difference of the zonally averaged ozone number density observed in July 2008, within $10\,^{\circ}$ latitude bins. OS, SMR and MIPAS relative to IRI are presented in first, second and third column, respectively. Upper and lower panels are 1D and 2D representation of this difference, respectively. The mean IRI ozone percentage error (precision) is also shown in grey colour in the upper panels.

ACE-FTS, and a positive bias of 10 to 20 % compared to MLS and SMILES (see their Fig. 7c). Investigating the influences of the local time sampling patterns and transport would require a comprehensive modelling study, as well as the processing
555  of a significantly bigger sample of the IRI data set allowing us to look for consistency and separate the different effects that contribute to the error.

## 4   Conclusions

In this study, we presented a technique to retrieve a new high along-track resolution IRA band and ozone data set. We first briefly presented the updated calibration scheme that is used on the OSIRIS IR imager level 1 data. Then, from the limb
560  radiance measurements, the optimal estimation method is used to retrieve the volume emission rate of $O_2(a^1\Delta_g)$ image by image. The implementation of the inversion includes a linearised scheme with a correction factor to account for the absorption process. The observed dayglow volume emission rate shows features such as a clear main layer and a secondary maximum, as well as finer structures along the orbit track. However, the nodding motion of the satellite occasionally leads to data gaps through the airglow layers.

From the retrieved volume emission rate of $O_2(a^1\Delta_g)$, ozone is derived based on the non-linear inversion of a photochemical kinetic model that describes the relationship between ozone and $O_2(a^1\Delta_g)$. The kinetic model is a slightly extended version of the one developed by Mlynczak et al. (1993), which includes mechanisms such as photo-dissociation of $O_2$ and $O_3$, solar excitation of $O_2$ and quenching of different excited stated of atomic and molecular oxygen.

However, the validity of the photochemical equilibrium assumption, which is essential when using the kinetic model to retrieve ozone, is questionable for a large portion of the IRI measurements that are located near the night-day terminator, especially near the equator above 75 km. To manage this issue we apply a novel approach where we integrate the uncertainty caused by this effect directly into the retrieval by increasing the measurement uncertainty according to divergence from the equilibrium level. This equilibrium index is assessed based on the time after local sunrise and the total lifetime of $O_2(a^1\Delta_g)$ as a function of altitude. As a result of such modification on the measurement uncertainty, the regions where the equilibrium assumption is far from valid are associated with low measurement response and high estimated error, and are thus screened out for further analysis, while other regions remain sensitive to the measurement.

Finally, the daytime ozone density retrieved from a test sample of IRI measurements made between November 2007 and October 2008 is compared to ozone products collected from external instruments, namely SMR, OS and MIPAS. The comparison of the coincident profiles of IRI, OS and SMR corresponding to an arbitrarily selected orbit shows that they merge rather well, although they do not cover the same altitude ranges. The comparison also demonstrates the advantage of the high sampling rate of IRI, which implies that, when averaging large samples, the random error can be greatly reduced. Zonally averaged monthly mean profiles give us an overall image of the inferred global distribution of ozone. It can be seen that IRI ozone data set is capable of reproducing the general seasonal and latitudinal distributions in the mesosphere - lower thermosphere, as it is shown in MIPAS and SMR in the same year. The relative difference between IRI and other instruments shows that IRI has a positive bias of up to 25% below 75 km, and up to 50% in some regions above. We think that this bias mostly comes from the uncertainty in the photochemical model, the time delay of the measured 1.27 μm signal in response to the changes in $O_3$ and the absolute calibration process of the limb radiance data but some may still be due to differences in the exact solar illumination conditions of the observations.

Overall, this study has demonstrated the technique of retrieving ozone density from the $O_2(a^1\Delta_g)$ limb radiance measurements from the IR imager on board the Odin satellite. The inter-comparisons with independent ozone data sets show that such a technique can be further applied to all IRI limb radiance data throughout the 19 years of the mission to date, leading to a new, long-term, high resolution ozone data set in the middle atmosphere.

*Data availability.* The data will be available on our ftp server odin-osiris.usask.ca on special request. More information is available at https://research-groups.usask.ca/osiris/

## Appendix A: Appended tables

**Table A1.** Reactions and their coefficients included in the photochemical model

| Symbol in Fig.5 | Reaction | Rate coefficient (in molecule cm s units) | Efficiency | Reference |
|---|---|---|---|---|
| $J_H$ | $O_3 + h\nu \to O_2(a^1\Delta_g) + O(^1D)$ | Vertical profile | 0.9 | JPL |
| $J_{src}$ | $O_2 + h\nu \to O(^3P) + O(^1D)$ | Vertical profile | | JPL |
| $J_{L\alpha}$ | $O_2 + h\nu \to O(^3P) + O(^1D)$ | Vertical profile | 0.44 | JPL |
| $g_A$ | $O_2 + h\nu \to O_2(b^1\Sigma_g^+, v=0)$ | Vertical profile | | HITRAN |
| $g_B$ | $O_2 + h\nu \to O_2(b^1\Sigma_g^+, v=1)$ | Vertical profile | | HITRAN |
| $g_{IRA}$ | $O_2 + h\nu \to O_2(a^1\Delta_g)$ | Vertical profile | | HITRAN |
| $A_1$ | $O(^1D) \to O + h\nu(\lambda=630\,\text{nm})$ | $6.81 \times 10^{-3}$ | | a |
| $A_2$ | $O_2(b^1\Sigma_g^+, v=0) \to O_2 + h\nu(\lambda=762\,\text{nm})$ | $8.34 \times 10^{-2}$ | | a |
| $A_3$ | $O_2(b^1\Sigma_g^+, v=1) \to O_2(b^1\Sigma_g^+, v=1) + h\nu(\lambda=771\,\text{nm})$ | $7.2 \times 10^{-2}$ | | a |
| $A_4$ | $O_2(a^1\Delta_g) \to O_2 + h\nu(\lambda=1.24\,\mu\text{m})$ | $2.26 \times 10^{-4}$ | | a |
| $Q_1$ | $O(^1D) + N_2 \to O(^3P) + N_2$ | $2.15 \times 10^{-11} \times \exp(-110/T)$ | | JPL |
| $Q_{1a}$ | $O(^1D) + O_2 \to O(^3P) + O_2(b^1\Sigma_g^+, v=1)$ | $3.3 \times 10^{-11} \times \exp(-55/T)$ | 0.8 | JPL, a |
| $Q_{1b}$ | $O(^1D) + O_2 \to O(^3P) + O_2(b^1\Sigma_g^+, v=0)$ | $3.3 \times 10^{-11} \times \exp(-55/T)$ | 0.2 | JPL, a |
| $Q_{2a}$ | $O_2(b^1\Sigma_g^+, v=1) + O_2$ $\to O_2(X^3\Sigma_g^-, v=1) + O_2(b^1\Sigma_g^+, v=0)$ | $2.2 \times 10^{-11} \times \exp(-115/T)$ | | a |
| $Q_{2a}$ | $O_2(b^1\Sigma_g^+, v=1) + O(^3P) \to O_2 + O(^3P)$ | $4.5 \times 10^{-12}$ | | a |
| $Q_{2a}$ | $O_2(b^1\Sigma_g^+, v=1) + O_3 \to 2O_2 + O(^3P)$ | $3 \times 10^{-10}$ | | a |
| $Q_{2a}$ | $O_2(b^1\Sigma_g^+, v=1) + N_2 \to N_2 + O_2(b^1\Sigma_g^+, v=0)$ | $7 \times 10^{-13}$ | | a |
| $Q_{2b}$ | $O_2(b^1\Sigma_g^+, v=0) + N_2 \to O_2(a^1\Delta_g) + N_2$ | $2.1 \times 10^{-15}$ | | JPL |
| $Q_{2b}$ | $O_2(b^1\Sigma_g^+, v=0) + O_2 \to O_2(a^1\Delta_g) + O_2$ | $3.9 \times 10^{-17}$ | | JPL |
| $Q_{2b}$ | $O_2(b^1\Sigma_g^+, v=0) + O \to O_2(a^1\Delta_g) + O$ | $8 \times 10^{-14}$ | | JPL |
| $Q_{2b}$ | $O_2(b^1\Sigma_g^+, v=0) + O_3 \to O_2(a^1\Delta_g) + O_3$ | $2.2 \times 10^{-11}$ | | JPL |
| $Q_{2b}$ | $O_2(b^1\Sigma_g^+, v=0) + CO_2 \to O_2(a^1\Delta_g) + CO_2$ | $4.2 \times 10^{-13}$ | | JPL |
| $Q_3$ | $O_2(a^1\Delta_g) + O_2 \to 2O_2$ | $3.6 \times 10^{-18} \times \exp(-220/T)$ | | JPL |
| $Q_3$ | $O_2(a^1\Delta_g) + N_2 \to O_2 + N_2$ | $1 \times 10^{-20}$ | | JPL |
| $Q_3$ | $O_2(a^1\Delta_g) + O \to O_2 + O$ | $2 \times 10^{-16}$ | | JPL |
| $Q_3$ | $O_2(a^1\Delta_g) + O_3 \to O_2 + O_3$ | $5.2 \times 10^{-11} \times \exp(2840/T)$ | | JPL |
| Barth | $2O + M \to O_2^* + M$ | $4.7 \times 10^{-33} \times \exp(300/T)$ | see footnote | b |
| Barth | $O_2^* + O, O_2, N_2 \to \text{allproducts}$ | see footnote | | b |
| Barth | $O_2^* + O_2 \to O_2 + O_2(b^1\Sigma_g^+)$ | see footnote | see footnote | b |

a: See reference list in Yankovsky et al. (2016)

b: Empirical quenching coefficients are introduced. In accordance with notation in McDade et al. (1986), $C^{O_2} = 6.6$, $C^O = 19$

JPL: See reference list in the JPL Publication 10-10 (Burkholder et al., 2015)

HITRAN: Gordon et al. (2017)

**Table A2.** A list of acronyms that have been used in the paper

| Acronym | Full spelling |
| --- | --- |
| HALOE | Halogen Occultation Experiment |
| ACE-FTS | tmospheric Chemistry Experiment - Fourier Transform Spectrometer |
| SOFIE | Solar Occultation for Ice Experiment |
| GOMOS | Global Ozone Monitoring by Occultation of Stars |
| SABER | Sounding of the Atmosphere using Broadband Emission Radiometry |
| MIPAS | Michelson Interferometer for Passive Atmospheric Sounding |
| SMR | Sub-Millimetre Radiometer |
| SME | Solar Mesosphere Explorer |
| SCIAMACHY | SCanning Imaging Absorption SpectroMeter for Atmospheric CHartographY |
| OS | Optical Spectrograph |
| IRI | Infrared Imager |
| OSIRIS | Optical Spectrograph and InfraRed Imaging System |
| METEORS | Mesosphere-Thermosphere Emissions for Ozone Remote Sensing |
| MLT | Mesosphere and lower thermosphere |
| OEM | Optimal estimation method |
| MAP | Maximum a posteriori |
| AVK | Averaging kernel |
| MR | Measurement response |
| FWHM | Full Width Half Maximum |
| SZA | Solar zenith angle |
| CMAM | Canadian Middle Atmosphere Model |
| MSIS | Mass Spectrometer Incoherent Scatter |
| ECMWF | European Centre for Medium-Range Weather Forecasts |
| VMR | Volume mixing ratio |
| IRA band | InfraRed Atmospheric band |
| A band | Atmospheric band |

*Author contributions.* The main author has prepared all the calculations and figures, the University of Saskatchewan authors have produced the calibrated IRI data and written Sect. 2.1. All authors have contributed to the discussions.

*Competing interests.* The authors declare that they have no competing interests.

*Acknowledgements.* We are thankful to the other instrument teams for access to their datasets. The Atmospheric Chemistry Experiment (ACE), also known as SCISAT, is a Canadian-led mission mainly supported by the Canadian Space Agency. Odin is a Swedish-led satellite project funded jointly by the Swedish National Space Agency (SNSA), the Canadian Space Agency (CSA), the National Technology Agency of Finland (Tekes), and the Centre National d'Etudes Spatiales (CNES) in France. Odin is also part of the ESA's third party mission programme. We also thank ESA and IMK for access to the MIPAS dataset.

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
