# Peer review of "Retrieval of daytime mesospheric ozone using OSIRIS observation of $O_2(a^1\Delta_g)$ emission"

_Atmospheric Measurement Techniques, 2020_

## Referee Comment (RC1) · Anonymous Referee #1 · 20 May 2020

General comments:

This manuscript presents a generally well written study on a new MLT ozone data set retrieved from O2 IR A-band airglow emission measurements with the IR imager on the Odin satellite. The calibration of the IRI measurements is described, as is the retrieval approach and a first comparison with independent satellite measurements. The study presents an interesting and relevant contribution to the field and should eventually be published. I ask the authors to address the following comments, many of which are really minor.

I have two general comments:

1. The section on the calibration of the IRI measurements should be more specific and

detailed. I guess this paper will be THE paper on the calibration of these measurements and will be used as a a reference for future papers. The description is not detailed enough to understand the details and to reproduce the individual steps. I'm not asking for every little detail to be explained, but more information on the vague parts should be provided (see also the specific comments below).

2. The agreement of the IRI O3 retrievals with some of the other data sets is not very good or rather poor at some altitudes, latitudes and/or time of the year. The authors offer different explanations for these differences, but issues with the photochemical model are not discussed. I think this is an obvious candidate to investigate. I'm not asking for new analyses etc. but suggest mentioning that the model may be an issue here and may / will be tested in a future study. In my opinion the data sets (IRI and co-located SMR measurements are a unique opportunity to test and improve the photochemical model.

Specific comments:

Title: "OSIRIS observation" –> "OSIRIS observations"?

Line 12: "19 years-long mission" -> "19-year mission" ?

Line 38: "affect the inferred ozone distribution, especially whose lifetime is comparable to the transport timescales."

Something is missing / wrong here. Please correct

Line 54: "Degenstein et al. (2005b)"

I suggest changing the order of the papers in the reference list such that Degenstein et al. (2005a) is cited first.

Line 64: "sample .. have been processed." -> "sample .. has been processed."

Line 70: "we also include MIPAS and ACE-FTS ozone profiles, measurements retrieved from other satellites"

This is only a really minor thing, but "measurements retrieved" sounds somewhat strange. I tend to associate "measurements" with the initial radiance spectra measurements. Perhaps you could write, e.g. ".. ozone profiles, i.e. data sets retrieved from measurements with instruments on other satellites". I leave it up to you to decide, whether you want to change this or not.

Line 83: "emissions with" -> " emissions with a" ?

Line 91: "dark current and electronic offset" -> "dark current and electronic offset correction"

Line 93: "version of (Bourassa, 2003)." -> "version of Bourassa (2003).", i.e. wrong cite command used.

Line 116: "The fitting process is a periodized"

Please explain what "periodized" means in this context. It is unclear to me.

Line 131: "The in-flight curves closely resemble the pre-flight curves with notable differences towards the edges of the arrays."

It would be interesting to show this comparison, because this paper will probably serve as a description of the calibration process also to be used for future studies.

Line 141: "The shape of the stray light is then extrapolated to lower tangent altitudes"

Please describe, how this is done. There are many different ways to extrapolate data.

Line 148: "or photons per second from a unit area"

from a unit area? I think it's photons passing through a unit area, right?

Same line: "The usual per nm wavelength dependence of the radiance"

This would then be "spectral radiance". The quantity with your units is simply "radiance"

Line 162: "The final reported error also incorporates the error in the pixel electronics

offset"

How is the final error determined based on the individual error components? This should be explained in more detail.

Figure 1, left panel, x-axis label: "Radiance"

Units missing.

Line 187: "The value of phi is relatively insensitive to the emission temperature."

Can this be quantified? If you have tested that it is relatively insensitive to T, you should be able to easily provide a rough quantitative estimate.

Caption, Figure 2: "Every two rows" -> "Every second row"?

Line 181: space missing in "B-band(688nm)"

Line 300: "are only sensitive below 90km or below." ?

Line 330: "Eq.9" -> "Eq. 9"

Caption, Figure 6: "Every two rows" -> "Every second row"?

Caption, Figure 7: "scaled with their corresponding a priori profiles." -> "divided by the corresponding a priori profiles."

Line 347: "the 20 years data" -> "20 year data set"

Line 386: "the thermal emission line of ozone"

Line? It's many, many lines, right?

Line 388: "Van Der A" -> "Van der A"

Line 406: "SMR ozone measures from" -> "SMR provides/measures ozone from"

Figure 8, IRI data: Why is O3 negative over such an extended altitude range? It would be good to discuss potential reasons in more detail. What about problems with

the photochemical model? Please also mention, whether the VERs are also already negative in these regions.

Line 425: "every 20th orbits .. have" -> "every 20th orbit .. has"

Line 343: "and therefore blanked out in Fig. 9."

Please check grammar of this sentence. Something is missing here.

Line 437: "MIPAS observes a deeper trough in the winter hemisphere as in IRI and SMR data, but a relatively even distribution in the MLT region."

I can't really see that in the figure. What does "deeper" refer to here? The ozone values or altitude? This is not clear.

Lines 441 following: The authors discuss differences in SZA as a cause for the differences between the different datasets. This is certainly a possible reason, at least for part of the differences. But are the differences between the data sets consistent with the diurnal variation of O3 and the different SZAs of the measurement shown in Fig. 10? This could be easily addressed qualitatively.

Figure 11, left panel: x-axis label is wrong -> "cm^-3"

Line 454 following: It should also be mentioned that the differences can be significantly larger at other latitudes.

Line 474: "Overall, IRI agrees very well with SMR"

Looking at Figures 9 and 12, I think this statement is not justified. Relative differences between the two data sets reach very large values, right?

Line 477 following: "The differences between IRI, ACE-FTS and MIPAS in Fig.12 may be explained .."

There may also be issues with the photochemical model used to retrieve O3. I think this should be explicitly mentioned. The dataset should be used in future studies to

[Figure]

attempt to improve the photochemical model used.

Figure 12: I suggest that negative values are more clearly indicated (e.g. in black). The current depiction makes it difficult to identify negative values.

Table A2: "ACE-FTS tmospheric"

---

## Referee Comment (RC2) · Anonymous Referee #2 · 28 May 2020

Dear Editor, Authors,

As my previous review was more a kind of formal/complete review rather than a quick review, it still prevails.

I have tried to access to the new/current verison but I couldn't. Also, yesterday was the deadline. Hence I am submitting again my previous review.

»» Previous Review ««

The manuscript does fall into the scope of AMT. It deals with a potential new data set of daytime mesospheric O3 and it is therefore very important. Eventually it can be sufficiently sound to be posted in its discussion forum but, in my opinion, not in its current form. I think it needs a substantial revision before it can be posted.

[Figure]

I would recommend to the authors the following actions:

The first part of the paper is well written, clear and well focused. This section, however, still requires some actions as:

1) Include a Table listing all error sources and their estimated values. It is true that in order to verify if the estimated systematic errors are plausible (for example that of the stray-light, particularly near 80 km when the atmospheric signal is very small but radiance for the bright region below might be important) one needs to perform a thorough validation. Hence, I propose to present that in the second part of the paper (see below).

2) To include a proper radiative transfer (i.e., do not assume the optically thin approach) for the 60-70 km region. Although the authors cite Degenstein (1999) (a Thesis work) as a support for the validity of this approach in that region, Mlynczak et al. (2007) state that "Below 70 km absorption of the O2(1D) emission by O2 itself begins to become important and the weak-line retrieval approach becomes invalid." I suggest that a) include full RT below 70 km, b) limit the data to 70 km, or c) estimate (quantitatively) the errors of that approach in the 60-70 km region.

3) If I have understood correctly, the 1.27 $\mu$m channel has not been calibrated in flight. It is just assumed that it behaves as other inflight-calibrated channels. I have not seen any error associated with this (lack of) calibration in the manuscript.

2nd part. Section 3.

This section is a mixture of a kind of a soft (descriptive, not rigorous) validation exercise together with some partial description of the behaviour of O3, incomplete from my point of view. In my view, none of the two aspects are shown with sufficiently sound scientific treatment.

For example, the current validation is only descriptive, a kind of hand-waving comparison, side by side figures, e.g., not showing differences, no co-location criteria has

been used (or at least not mentioned). Sentences like "The differences between IRI, ACE-FTS and MIPAS in Fig.12 may be explained by their sampling at different SZA and the underlying assumptions in their retrieval techniques." are very vague and little informative. If sampling is a cause, the differences should be looked at by restricting it. etc.

About the second aspect, the study is mainly descriptive and based on partial datasets and considering the O3 number density instead of the O3 vmr. Sentences like: "Thus such a monthly mean profile should be treated with caution since it may not necessarily well represent the spatial and temporal distribution of daytime ozone." again adds little information. If it does not represent the distribution, why then show it? The study on the "Monthly mean ozone" is based on 1 month of one year and of O3 density. I would suggest the authors to refer to other similar studies (e.g. Lopez-Puertas et al. 2018).

My recommendation for this section 3, would be to focus on a thorough, rigorous validation (following the standard guidelines, see some more comments below) and leave aside from this paper the kind of characterisation of O3 features. Maybe the authors would like to consider future papers as, e.g. the overall OSIRIS O3 data sets, or tackle comprehensive works, including or not other datasets, as seasonal/latitudinal variations, or local time and annual variations, etc.

Recommendations on validation:

1) It should be based on collocated data and, whenever possible, the same physical conditions, e.g., based on a coincidence criteria of time, spatial (latitude, longitude) and local time.

2) Compare the appropriate instruments. That is, there is no altitude overlap of IRI wrt SO. Hence I see no reason for including SO data.

- About MIPAS data, I suggest the author include a more appropriate reference, e.g., Lopez-Puertas et al. (2018). MIPAS middle atmosphere data ranges from ~20 km (not

5 km, Table 1) up to ∼100 km. BTW, I believe the authors have used only DAYTIME MI-PAS data (not stated explicitly in the manuscript). Mention which pressure/temperature is used (MSIS?, MIPAS?) to calculate O3 density.

- I would be inclined to not include ACE data. O3 shows a large diurnal variation in the mesosphere and ACE is always measuring at the terminator. Hence it is difficult to distinguish systematic differences inherent to the instruments from those due to the solar illumination. BTW, the authors should state early in the paper that the 1.27 mm emission has a radiative lifetime of approximately 75 min and does not provide a representative measure of ozone until 2–3 h after sunrise (Mlynczak et al., 2013). Has this fact been taken into account in the current comparison? This fact automatically should avoid to compare to ACE sunrise occultations.

- I am really missing a validation against SABER. In particular the O3 derived from the O2 1.27 $\mu$m channel. This is an instrument that uses the same technique and would therefore be very valuable.

3) Quantify the differences (of the co-located data) for the different seasons/latitudes/altitudes. That is, as Fig. 11, but including more altitudes/seasons and enough years of retrievals to make the statistic significant.

About the 2nd part, a description of the O3 characteristics should be presented in a different paper and I would recommend the authors to please cite other previous recent works about this.

Other minor points:

In general several figures are very small, particularly those with several panels, e.g. Fig. 3, 5, 9 and 11

Fig. 3 Could you show the solar local time?

a typo: earth -> Earth

Refs.  Lopez-Puertas M, García-Comas M, Funke B, et al.  MIPAS observations of ozone in the middle atmosphere.  Atmos Meas Tech.  2018;11(4):2187-2212. doi:10.5194/amt-11-2187-2018.

Mlynczak MG, Hunt LA, Mast JC, et al. Atomic oxygen in the mesosphere and lower thermosphere derived from SABER: Algorithm theoretical basis and measurement uncertainty. Journal of Geophysical Research. 2013;118(11):5724-5735.

---

## Author Comment (AC1) · 24 Aug 2020

General comments: This manuscript presents a generally well written study on a new MLT ozone data set retrieved from O2 IR A-band airglow emission measurements with the IR imager on the Odin satellite. The calibration of the IRI measurements is described, as is the retrieval approach and a first comparison with independent satellite measurements. The study presents an interesting and relevant contribution to the field and should eventually be published. I ask the authors to address the following comments, many of which are really minor. I have two general comments: 1. The section on the calibration of the IRI measurements should be more specific and detailed. I guess this paper will be THE paper on the calibration of these measurements and will be used as a a reference for future papers. The description

is not detailed enough to understand the details and to reproduce the individual steps. I'm not asking for every little detail to be explained, but more information on the vague parts should be provided (see also the specific comments below). A more detailed description on calibration will be included in another paper, which is aimed for submission in several months by Saskatoon authors. 2. The agreement of the IRI O3 retrievals with some of the other data sets is not very good or rather poor at some altitudes, latitudes and/or time of the year. The authors offer different explanations for these differences, but issues with the photochemical model are not discussed. I think this is an obvious candidate to investigate. I'm not asking for new analyses etc. but suggest mentioning that the model may be an issue here and may / will be tested in a future study. In my opinion the data sets (IRI and co-located SMR measurements are a unique opportunity to test and improve the photochemical model. The issue with the uncertainty in the photochemical model is mentioned in the introduction section as well as in the ozone retrieval section. But this will be re-emphasized again in the result/discussion section in the updated version of the manuscript. Indeed, IRI and SMR measurements are a unique opportunity to test and improve the photochemical model in a future study. In the course of this study we noted some instability in the mesospheric part of the SMR profiles. While the average profiles are fine in order to properly tune the model individual profiles of good quality are needed. We therefore reserve this for a later. We would like to thank referee #1 for the valuable input to help us improve the manuscript. Specific comments: Title: "OSIRIS observation" –> "OSIRIS observations"? Corrected in the revised version. Line 12: "19 years-long mission" -> "19-year mission" ? Corrected in the revised version. Line 38: "affect the inferred ozone distribution, especially whose lifetime is comparable to the transport timescales." Something is missing / wrong here. Please correct. Corrected in the revised version as "Furthermore, the photochemical timescales of the airglow species critically affect the inferred ozone distribution, especially species whose lifetime is comparable to the transport timescales." Line 54: "Degenstein et al. (2005b)" I suggest changing the order of the papers in the reference list such that Degenstein et al.

Interactive
comment

(2005a) is cited first. Degenstein et al. (2005a) is cited in line 26 before Degenstein et al. (2005b) is cited in line 54. Line 64: "sample .. have been processed." -> "sample .. has been processed." Corrected in the revised version. Line 70: "we also include MIPAS and ACE-FTS ozone profiles, measurements retrieved from other satellites" This is only a really minor thing, but "measurements retrieved" sounds somewhat strange. I tend to associate "measurements" with the initial radiance spectra measurements. Perhaps you could write, e.g. ".. ozone profiles, i.e. data sets retrieved from measurements with instruments on other satellites". I leave it up to you to decide, whether you want to change this or not. Corrected in the revised version. Line 83: "emissions with" -> " emissions with a" ? Corrected in the revised version. Line 91: "dark current and electronic offset" -> "dark current and electronic offset correction" Corrected in the revised version. Line 93: "version of (Bourassa, 2003)." -> "version of Bourassa (2003).", i.e. wrong cite command used. Corrected in the revised version. Line 116: "The fitting process is a periodized" Please explain what "periodized" means in this context. It is unclear to me. The fitting process is split into temporal chunks that span several calibration periods of the IR instrument (which are roughly every 50 orbits for the bulk of the mission). In this way, small changes in the calibration parameters can be tracked as the satellite instrument ages. This will be clarified in the upcoming paper Line 131: "The in-flight curves closely resemble the pre-flight curves with notable differences towards the edges of the arrays." It would be interesting to show this comparison, because this paper will probably serve as a description of the calibration process also to be used for future studies. This will be clarified in the upcoming paper. Line 141: "The shape of the stray light is then extrapolated to lower tangent altitudes" Please describe, how this is done. There are many different ways to extrapolate data. This is a big section with a lot of discussion, thus it will be left in the upcoming paper. Line 148: "or photons per second from a unit area" from a unit area? I think it's photons passing through a unit area, right? Corrected in the revised version. Same line: "The usual per nm wavelength dependence of the radiance" This would then be "spectral radiance". The quantity with your units is

simply "radiance" The usual unit of spectral radiance is not used in this formulation as the spectral information is effectively lost by integration of the signal across the passband. The radiance or brightness units are then photons per second. Line 162: "The final reported error also incorporates the error in the pixel electronics offset" How is the final error determined based on the individual error components? This should be explained in more detail. This will be clarified in the upcoming paper Figure 1, left panel, x-axis label: "Radiance" Units missing. Corrected in the revised version. Line 187: "The value of phi is relatively insensitive to the emission temperature." Can this be quantified? If you have tested that it is relatively insensitive to T, you should be able to easily provide a rough quantitative estimate. In the updated retrieval procedure, we have implemented the temperature dependent fraction of the optical filter overlapping the emission band, i.e. the value of phi, being temperature dependent, based on the temperature at the tangent point. The description is added to the revised version of the manuscript. Caption, Figure 2: "Every two rows" -> "Every second row"? Corrected in the revised version. Line 181: space missing in "B-band(688nm)" Corrected in the revised version. Line 300: "are only sensitive below 90km or below." ? Corrected in the revised version. Line 330: "Eq.9" -> "Eq. 9" Corrected in the revised version. Caption, Figure 6: "Every two rows" -> "Every second row"? Corrected in the revised version. Caption, Figure 7: "scaled with their corresponding a priori profiles." -> "divided by the corresponding a priori profiles." This figure is deleted. However, the updated/added figures which show the relative error are written 'error size relative to the individual a priori profiles' in the captions. Line 347: "the 20 years data" -> "20 year data set" Corrected in the revised version as 'the 20-year-dataset'. Line 386: "the thermal emission line of ozone" Line? It's many, many lines, right? No, it is a single line Line 388: "Van Der A" -> "Van der A" Corrected in the revised version Line 406: "SMR ozone measures from" -> "SMR provides/measures ozone from" Corrected in the revised version Figure 8, IRI data: Why is O3 negative over such an extended altitude range? It would be good to discuss potential reasons in more detail. What about problems with the photochemical model? Please also mention,

whether the VERs are also already negative in these regions. VERs are not negative but very low for this region. The main reason for such an extended region of negative values mainly comes from the fact that the photochemical equilibrium assumption is used in the model, while this assumption is hard to be considered valid in that region. On top of that, the model includes the contribution from O2 ground state to produce O2(1aDelta), therefore the inversion tries to force O3 to be very low, even negative, to overcompensate the low VERs being observed. In short, the steady state assumption will underestimate O3 concentration when this assumption is not valid. We have added this discussion in the revised manuscript, as well as a novel approach to address this issue in an updated procedure for reprocessing IRI O3 data (see the newly added Sect. 2.3.3.) Line 425: "every 20th orbits .. have" -> "every 20th orbit .. has" Corrected in the revised version Line 343: "and therefore blanked out in Fig. 9." Please check grammar of this sentence. Something is missing here. (perhaps Line 434) The section and figure have been rewritten in the revised version therefore this comment is no longer relevant. Line 437: "MIPAS observes a deeper trough in the winter hemisphere as in IRI and SMR data, but a relatively even distribution in the MLT region." I can't really see that in the figure. What does "deeper" refer to here? The ozone values or altitude? This is not clear. The section and figures have been rewritten in the revised version therefore this comment is no longer relevant. Lines 441 following: The authors discuss differences in SZA as a cause for the differences between the different datasets. This is certainly a possible reason, at least for part of the differences. But are the differences between the data sets consistent with the diurnal variation of O3 and the different SZAs of the measurement shown in Fig. 10? This could be easily addressed qualitatively. This figure and the discussions about differences in SZA sampling are no longer in the revised manuscript, as the reprocessed IRI data show significantly closer to MIPAS zonal mean data compared to the version before. Figure 11, left panel: x-axis label is wrong -> "cmËĘ-3" This figure is no longer in the revised manuscript. Line 454 following: It should also be mentioned that the differences can be significantly larger at other latitudes. We have expanded

the difference figure to include all overlapping latitude bins (see Fig. 14 in the revised version), excluding ACE-FTS as referee #2 provided the reason not to include this dataset in the paper. Line 474: "Overall, IRI agrees very well with SMR" Looking at Figures 9 and 12, I think this statement is not justified. Relative differences between the two data sets reach very large values, right? With the newly added Fig. 14 in the revised version, it is shown that the updated IRI data has generally 20-50% positive bias compared to SMR and MIPAS. Line 477 following: "The differences between IRI, ACE-FTS and MIPAS in Fig.12 may be explained .." There may also be issues with the photochemical model used to retrieve O3. I think this should be explicitly mentioned. The dataset should be used in future studies to attempt to improve the photochemical model used. We acknowledge this comment. This is emphasized in the revised version. Figure 12: I suggest that negative values are more clearly indicated (e.g. in black). The current depiction makes it difficult to identify negative values. After the IRI data being reprocessed with the modification in measurement uncertainty scaling with the equilibrium index, the negative values in such plots are mostly replaced by a priori value with low measurement response. Data with low measurement response and low equilibrium index are filtered out before making an averaging profile. Table A2: "ACE-FTS tmospheric" Thank you for pointing out the typo. However, ACE-FTS will be removed from this paper for ozone comparison as pointed out by referee #2.

Please also note the supplement to this comment:
https://amt.copernicus.org/preprints/amt-2020-56/amt-2020-56-AC1-supplement.pdf

---

## Author Comment (AC2) · 24 Aug 2020

Ref #2 A general reply to Ref #2: We do not intend the main focus of the paper to be a complete and comprehensive validation study of the new data set of mesospheric O3. Rather in this paper we focus on presenting the retrieval technique, and the utility of the OSIRIS limb emission profiles as a sample data set for this type of high resolution mesospheric O3 data product. This primary goal is mentioned in the end of Sect.1 and beginning of Sect.3. Indeed, if the entire ∼20 year Odin-OSIRIS data is processed, it would comprise a very valuable data set as the reviewer acknowledges. The processing of the entire 20 year OSIRIS data set with the new instrument corrections and using the inversion technique to the Level 2 product is a substantial computational undertaking and must be addressed in future work beyond this first paper. We would

like to thank to referee #2 by raising several important discussion questions that have helped us to improve the manuscript and the data processing method, namely the equilibrium assumption issue accompanying 1.27 $\mu$m emission being the proxy of O3, and the absorption corrections on the VER retrieval. The manuscript does fall into the scope of AMT. It deals with a potential new data set of daytime mesospheric O3 and it is therefore very important. Eventually it can be sufficiently sound to be posted in its discussion forum but, in my opinion, not in its current form. I think it needs a substantial revision before it can be posted. I would recommend to the authors the following actions: The first part of the paper is well written, clear and well focused. This section, however, still requires some actions as: 1) Include a Table listing all error sources and their estimated values. It is true that in order to verify if the estimated systematic errors are plausible (for example that of the stray-light, particularly near 80 km when the atmospheric signal is very small but radiance for the bright region below might be important) one needs to perform a thorough validation. Hence, I propose to present that in the second part of the paper (see below). We agree that a table summarising all error sources and their estimated values would improve the paper. However, quantification some of the error sources can only be done after more data being processed, or require a comprehensive modelling study such as in Zhu et al. (2007), thus it will be left to the future validation study. We have however provided a table of possible error sources, quantifying those that can be quantified and estimation others at the end of Sect. 2.

2) To include a proper radiative transfer (i.e., do not assume the optically thin approach) for the 60-70 km region. Although the authors cite Degenstein (1999) (a Thesis work) as a support for the validity of this approach in that region, Mlynczak et al. (2007) state that "Below 70 km absorption of the O2(1D) emission by O2 itself begins to become important and the weak-line retrieval approach becomes invalid." I suggest that a) include full RT below 70 km, b) limit the data to 70 km, or c) estimate (quantitatively) the errors of that approach in the 60-70 km region. We acknowledge this limitation. We have reprocessed the VER retrieval by introducing an absorption correction factor to

address this issue. The correction factor is calculated based on the tangent pressure (see the revised Sect. 2.2). The changes on the resulting VERs are less than 10% in the 60-70 km region, as shown in the figure below.

The figure shows the correction factor along the tangent path compared to the optically thin case along the tangent path. Colours correspond to different tangent heights of the paths in the unit of km. As can be seen from the red line representing line-of-sight tangent of 60km, the absorption would result in approximately 10% underestimation for the band intensity for emission near the tangent point. Furthermore, this factor is formatted in a pressure grid and applied to the retrieval scheme according to the pressure level at the tangent point. 3) If I have understood correctly, the 1.27 $\mu$m channel has not been calibrated in flight. It is just assumed that it behaves as other inflight-calibrated channels. I have not seen any error associated with this (lack of) calibration in the manuscript. We acknowledge this comment. Odin carries no optical calibration sources. As mentioned in Sect. 2.1.4, the absolute calibration relies on the pre-launch value, and for the 1.53 $\mu$m channel this has been confirmed in flight as indicated in the paper. As at this point, assessment of the uncertainty due to the in-flight changes in the absolute calibration is difficult. The long term stability of this channel will however need to be assessed when the full 20 years of data is processed. Thus the characterisation of this type of error source will be addressed in the future study. However, we have included this in the error source list as suggested in point 1) in the revised manuscript.

2nd part. Section 3. This section is a mixture of a kind of a soft (descriptive, not rigorous) validation exercise together with some partial description of the behaviour of O3, incomplete from my point of view. In my view, none of the two aspects are shown with sufficiently sound scientific treatment. We would again like to emphasise that we do not intend this paper to be a comprehensive validation study of a new data product. We acknowledge the reviewer's point that the comparative statements regarding the measurements by the various instruments are somewhat soft, yet we believe these

are useful comparisons (not validation) to show the fidelity of the retrieval technique. We agree with the reviewer that a comprehensive validation that uses the entire Odin-OSIRIS data set would indeed be valuable and we hope to be able to address this in a future paper. For example, the current validation is only descriptive, a kind of hand-waving comparison, side by side figures, e.g., not showing differences, no co-location criteria has been used (or at least not mentioned). Sentences like "The differences between IRI, ACE-FTS and MIPAS in Fig.12 may be explained by their sampling at different SZA and the underlying assumptions in their retrieval techniques." are very vague and little informative. If sampling is a cause, the differences should be looked at by restricting it. Etc. We agree that this comparison is descriptive. We mainly intend to demonstrate an Odin internal comparison of their co-incident profiles, as they are based on different physical measurement techniques and overlapping. Yet, these side-by-side figures show that the ozone profiles from three independent Odin-borne instruments complement each other well. Showing the differences between the coincident profiles would not be particularly informative in this case, due to their limited overlapping altitude range. Regarding the comparison of zonal mean monthly average distribution with other instruments, an updated difference plot is provided in Fig. 14 including several latitude bands in order to make it more complete. We would like to point out again that, as stated in Sect. 3, only a test sample of 5% of all the measurements collected during a one-year period has been processed for this study. This information has been also added to the introduction section in the revised version. About the second aspect, the study is mainly descriptive and based on partial datasets and considering the O3 number density instead of the O3 vmr. Sentences like: "Thus such a monthly mean profile should be treated with caution since it may not necessarily well represent the spatial and temporal distribution of daytime ozone." again adds little information. If it does not represent the distribution, why then show it? The study on the "Monthly mean ozone" is based on 1 month of one year and of O3 density. I would suggest the authors to refer to other similar studies (e.g. Lopez-Puertas et al. 2018). We use number density mainly as it is the natural unit for the IRI measurements. Moreover,

for SMR O3 at higher altitudes, the lines are mainly Doppler broadened rather than pressure broadened, thus the natural unit is closer to number density rather than VMR. Of course, IRI can be converted to VMR, by introducing external data such as MSIS, as shown in a small portion of Fig. 11. As Smith et al. 2003 mentioned, the different background density to derive VMR may introduce additional uncertainty between O3 profiles from different instruments, we would like to avoid introducing external data. We have addressed this issue in the revised version. Fortunately, thanks to the updated scheme of ozone retrieval, the zonal mean monthly average ozone demonstrated in Fig. 12 and Fig. 13 agree much better with MIPAS than those in the previous version of the manuscript. We agree that the study is descriptive in regard to the behaviour of O3, indeed based on what we can tell from the limited amount of data we have at the moment. We intended to confirm that IRI data is able to reproduce a general pattern of the O3 distribution in the MLT. My recommendation for this section 3, would be to focus on a thorough, rigorous validation (following the standard guidelines, see some more comments below) and leave aside from this paper the kind of characterisation of O3 features. Maybe the authors would like to consider future papers as, e.g. the overall OSIRIS O3 data sets, or tackle comprehensive works, including or not other datasets, as seasonal/latitudinal variations, or local time and annual variations, etc. We want to emphasise again that a comprehensive validation study is not the focus of this paper. We acknowledge that a rigorous validation study is indeed valuable for a future study, after the ∼20 years data will be processed. Section 3 is intended to provide a first-hand comparison in order to demonstrate the fidelity of the retrieval technique described in Sect. 2, which is our primary focus in this paper. We have described our purpose more clearly in the revised version. Recommendations on validation: 1) It should be based on collocated data and, whenever possible, the same physical conditions, e.g., based on a coincidence criteria of time, spatial (latitude, longitude) and local time. This issue can be addressed providing that more IRI data are processed and a sufficient amount of coincidence profiles between MIPAS and IRI are found. However, as already explained above, this can not be done at this stage. 2) Compare the appropriate

instruments. That is, there is no altitude overlap of IRI wrt SO. Hence I see no reason for including SO data. After the reprocessing of the IRI data with a correction on absorption, data can reach as low as 40km (50km after the last 10 grids were removed to avoid edge effect coming from the retrieval), thus IRI and OS has at least 10km of overlapping. Moreover, it is the first time to demonstrate how these three ozone data sets from Odin complement each other so well, despite their intrinsically different underlying physics in terms of measurement techniques. This also shows for the broader scientific community how Odin can cover a large part of the atmosphere using its different instruments. Thus, we do not want to completely remove the OS data set from this study, particularly Fig. 11.

- About MIPAS data, I suggest the author include a more appropriate reference, e.g., Lopez-Puertas et al. (2018). MIPAS middle atmosphere data ranges from ∼20 km (not 5 km, Table 1) up to ∼100 km. BTW, I believe the authors have used only DAYTIME MIPAS data (not stated explicitly in the manuscript). Mention which pressure/temperature is used (MSIS?, MIPAS?) to calculate O3 density. We acknowledge the reference. MIPAS night time ozone is screened out as described in line 442. To calculate the density, we use pressure and temperature measured by MIPAS. We have provided this information in the revised version. - I would be inclined to not include ACE data. O3 shows a large diurnal variation in the mesosphere and ACE is always measuring at the terminator. Hence it is difficult to distinguish systematic differences inherent to the instruments from those due to the solar illumination. BTW, the authors should state early in the paper that the 1.27 mm emission has a radiative lifetime of approximately 75 min and does not provide a representative measure of ozone until 2–3 h after sunrise (Mlynczak et al., 2013). Has this fact been taken into account in the current comparison? This fact automatically should avoid to compare to ACE sunrise occultations. We agree that, for ACE data, it is difficult to distinguish the differences due the solar illumination from those inherent to the instrument. Thus we have removed ACE data for this study. Regarding the long radiative lifetime of the 1.27 $\mu$m emission, we have added this information early in the revised version of

the manuscript. In addition, we address this issue in an updated inversion process of ozone by increasing the uncertainty matrix of the VER, based on an 'equilibrium index', Index = 1-exp(-t/ðİŻŢ), where ðİŻŢ is the total lifetime (ðİŻŢ = 1/(A+Q)) of the 1.27 $\mu$m emission, being the combination of the emission lifetime (1/A) and the quenching lifetime (1/Q) at a given altitude. This index indicates how far to the photochemical equilibrium state at a given altitude and time after the local sunrise. The detailed descriptions are provided in the newly added Sect. 2.3.3. In short, the effect of the newly introduced 'equilibrium index' in the O3 retrieval will suppress the underestimation of O3 where the steady state assumption is invalid, and thus these regions will be filtered out by low measurement responses before making comparisons with other ozone datasets. - I am really missing a validation against SABER. In particular the O3 derived from the O2 1.27 $\mu$m channel. This is an instrument that uses the same technique and would therefore be very valuable. Indeed it will be very valuable to compare SABER 1.27 O3 to IRI O3 in a future study, when the full data set will be processed and we will be able to carry out a validation study rigorously based on the comparison of coincident profiles. However, measurements from SABER drift in local time, unlike all other instruments included in this paper. For this reason, we have decided not to include SABER in this first study. 3) Quantify the differences (of the co-located data) for the different seasons/latitudes/altitudes. That is, as Fig. 11, but including more altitudes/seasons and enough years of retrievals to make the statistic significant. A new figure including differences of more latitudes is added, see Fig. 14. The comparisons between IRI and OS, SMR are always coincident. Based on the available IRI one-year-data set, it is difficult to have a statistically significant number of coincident profiles with MIPAS. As mentioned earlier, the focus of this study is on the retrieval technique. Processing a statistically significant amount of data for several years would require substantial computation undertaking and thus this should be left for future studies. About the 2nd part, a description of the O3 characteristics should be presented in a different paper and I would recommend the authors to please cite other previous recent works about this. Other minor points: In general several many

figures are very small, particularly those with several panels, e.g. Fig. 3, 5, 9 and 11 It is revised in the new version of manuscript. Fig. 3 Could you show the solar local time? Solar local time for Odin is relatively constant (6h,18h). Instead, we have added the time after the local sunrise axis in the figure. a typo: earth -> Earth It is revised in the new version of manuscript.

Please also note the supplement to this comment:
https://amt.copernicus.org/preprints/amt-2020-56/amt-2020-56-AC2-supplement.pdf

---

## Author Response (AR2)

REF #1

General comments:

Overall the authors did a good job responding to the points raised in my earlier review and I also find the new parts of the manuscript interesting and relevant. I have some more specific comments that I ask the authors to consider, most of them very minor. One more general point is the treatment of the measurements made close to local sunset. The equilibrium calculations are only done for the morning sector, i.e. measurements after local sunrise. But the effects will also become important before sunset, right? IRI also takes measurements close to sunset, but it's not clear, whether these were processed or not or treated in a special way? I apologize if I missed something here.

We would like to thank referee #1 for checking all the details and making suggestions for a further improvement of the manuscript. The problem of the equilibrium assumption is of course always present manifested as a time delay, but the largest effect is at sunrise. Please see the detailed explanation below.

Specific comments:

Line 13: 'However, the IRI ozone data set is consistent with the compared data set about the overall atmospheric distribution of ozone.'

This sentence is somewhat odd. I suggest replacing, 'about' by 'regarding' or 'in terms of', e.g..

We have replaced 'about' with 'in terms of' in the revised version.

Line 14: 'We attribute these differences TO'

Updated in the revised version.

Line 15: 'This implies that'

It's unclear what 'This' refers to and the logical connection between the sentences is unclear. I guess you don't really intend to argue that the retrieval should be applied to the entire data set, because of the calibration issues and potential problems with the photochemical model?

The order of the previous two sentences are switched to maintain the logical connection with this sentence in line 15.

Line 43: 'especially OF the species' ?

Updated in the revised version.

Same line: ´whose lifetime are´ -> ´whose lifetime is´ or ´whose lifetimes are´ ?
Updated in the revised version.
Line 81: ´And agree, despite their intrinsically´

This is not a complete sentence.
We have deleted 'And agree' in the revised version .

Line 106: ´this small number .. have´ -> ´this small number .. has´
Updated in the revised version.

Line 156: The usual per nm wavelength dependence of the radiance, common in remote sensing observations of a spectrum, it is not used in this formulation as the measurement is an integral over the infrared band filter´

The quantity with the units you report is generally called ´radiance´. If a ´1/nm´ is added to the units, the corresponding quantity is generally called ´spectral radiance´. In this respect, calling your quantity ´radiance´ is absolutely fine.
Updated in the revised version.

Line 166: ´the error in the measured digital number of read-out electronics´

This is not well phrased and may be misunderstood. Please correct.
Updated in the revised version.

Fig. 1: Is there a simple reason, why the relative calibration errors changed with altitude?
The relative calibration errors increase toward the edges of the IRI photodetector array. This is because the relative calibration is based on brightness comparisons between neighboring photodiodes. As we get further and further away from the central photodiodes, the error compounds and thus increases.

In the plot in question, it is the photodiodes at the lower numbered edge that are measuring the highest altitudes, thus they have more RC error than the other photodiodes in that image.

Line 194: ´((´ and ´))´
Corrected in the revised version.

Caption Fig. 2: I suggest deleting ´size´ in ´error size´
Corrected in the revised version.

Line 226: ´This, because´

Something is missing here.
It is changed to 'This is because…' in the revised version

Line 238: ´returning back´ -> ´returns back´
Updated in the revised version

Caption Fig. 4: I suggest deleting ´size´ in ´error size´
Corrected in the revised version.

Caption Fig. 4: ´, with 'nan' labels the sunrise is absence at the summer pole´

Please improve grammar.
Changed to '... where  'nan' indicates a location near the summer pole where sunrise is absent. '

Same line: ´the cross section´

It's simply ´vertical profiles´, not cross sections, right?
Updated in the revised version.

Line 260: ´will lead to under-estimate ozone´ -> ´will lead to an underestimation of ozone´
Updated in the revised version.

In addition: I'm wondering, whether such an approach will always underestimate ozone. You are arguing for the morning sector, i.e. after local sunrise, but what about the evening sector (before sunset). If I understand correctly, your argument for the near-sunrise measurements is, that the $O_2(a_1\Delta)$ state has a relatively long lifetime (74 min.) and when it is formed (mainly by $O_3$ photolysis) after sunrise, the emission at a certain time does not reflect the concentration of $O_2(a_1\Delta)$ at this time. A simplified approach would then underestimate ozone, I agree. But what about the evening sector? $O_3$ will increase around sunset, because $O_3$-photolysis gets smaller and smaller. The emission, however, will reflect the $O_2(a_1\Delta)$ production at an earlier point in time. It appears

that ozone will also be underestimated, if these effects are not taken into account, right?

I suggest explaining the difference between measurements close to local sunrise and sunset in a few sentences. And I think your approach with the equilibrium level does only work for the measurements after sunrise, right? Perhaps this should be mentioned as well.
OSIRIS will also make (close to) sunset measurements of course. I think it's not mentioned in the paper that these measurements are not used (perhaps I missed it). This seems an important point, I think.
The approach employed allows us to deal with the "turn on" of the O2(1a Delta) production at sunrise.  It will not compensate for the time delay connected with changes in ozone throughout the day, here we will always have an extra source of uncertainty.  Looking at model results (Allan et al 1984) the major changes are post sunset or at least at solar zenith angle beyond the range used for the retrievals.  This has been clarified in the updated version.

Line 313: ´while the other photochemical sources are only sensitive below 90km´

The 'g_a-process' is also strongly dependent on SZA above 90 km, right, i.e. the statement is not fully correct.
It is corrected in the revised version.

Line 314: ´The Barth-type mechanism contributes very little and mainly between 100-110km´

Well, it's rather 90 - 100 km, right?
It is changed to 90-105 km in the revised version.

Line 329: ´x´ should be bold like in the equation below.
Updated in the revised version.

Line 353: ´such an assumption will lead to an under-estimation of the derived ozone.´

See my earlier comments on this aspect.
Refer to the earlier reply on this question.

Caption Fig. 7: ´A naive estimation´

I suggest mentioning briefly in what sense this estimation is naive.

'...assuming all collected measurements of O2(a1Delta_g) are in equilibrium state...' is added to the caption.

'... equilibrium index (see text)' is added to the caption.

Updated accordingly in the revised version.

Please improve grammar.
Has been changed to 'As the ratio t/tau takes the values of 1.6, 2.3, 3, 4 and 4.6,... '

Equation (20): is the power 8 intended? If yes, why was this power used?
The power 8 was chosen so that the area with low equilibrium index will result in a low measurement response. The following sentence is added after the equation to clarify that point: 'where the equilibrium index is raised to the power of 8 in order to force a sufficiently low measurement response in the relevant time and altitude ranges so that the affected data can be filtered out.'

Is it really 'no data'? This is difficult to see because of the color scale. Looking at the Fig. it seems some data is available.
Corrected as 'Nearly no data is available…' in the revised version.

Please improve grammar.
Updated in the revised version as '...is as large as 50% as seen in... '.

Line 400: Suggest replacing ´The detail investigation may´ by ´A detailed investigation of the reasons for these differences may ..´
Corrected in the updated version.

Line 401: ´detail modelling´ -> ´detailed modelling´

Corrected in the updated version.

Corrected in the updated version.

Corrected in the updated version.

Line 403: ´source´ -> ´sources´
Corrected in the updated version.

Table 1: ´ <20% below 90 km, >100% above 90 km´

This is a step from <20% to > 100% at 90 km! This doesn't really seem plausible?
Changed to 20 - 100% above 90 km. It comes from a modelling study by Yankovsky et. al (2016) where it is reported that the uncertainty exceeds 100% above 90km.

Line 449: ´the thermal emission line´

It's a band, not a line, right?
Yes it is a band -  Corrected in the text

Line 472: ´the agreement .. worsenS´
Corrected in the revised version.

Line 477: ´in an altitude-time series´

It's not really time series, right?
Changed to 'altitude-time plots' in the revised version.

Line 481: ´in both hemisphereS´
Corrected in the revised version.

Line 482: ´that THE IRI ozone ..´
Corrected in the revised version.

Line 484: ´while BEING relatively stable´
Corrected in the revised version.

Line 490: ´values exist in the regions between the secondary and primary ozone layer IN THE SMR DATA.´
Corrected in the revised version.

Caption Fig. 13: ´colour scales of 2D-histograms´ -> ´greyscales of 2D-histograms´

Corrected in the revised version.

Same line: ´The upper colour bar´ -> ´The upper greyscale´

Corrected in the revised version.

Line 506: ´of daytime measurementS´

Corrected in the revised version.

Line 511: ´with still more data NEAR the summer pole´

Corrected in the revised version.

Line 512: ´IRI loses most of the data above 70 km in the tropics, since they were mostly measured very close to the local sunrise.´

What about the measurements carried out close to sunset?

If the measurement part of the orbit is close to the local sunrise (slightly after), then the opposite node will be after the local sunset and thus no data will be available above 70 km. Please refer to the horizontal labels in Fig. 11, for example.

Line 527: ´may be THE main reasonS´

Corrected in the revised version.

Line 528: tidal effects are also non-negligible at altitudes lower than 90 km.

It is changed to 'tidal effects can be significant above 90 km' in the revised version.

Fig. 14, top left panel: I suggest reducing the size of the legend – or making it more transparent.

The figure is updated in the revised version.

Line 544: ´a clear a´ -> ´a clear´

Corrected in the revised version.

Line 562: ´do not measure over the same´ -> ´do not cover the same´

Corrected in the revised version.

Line 570: ´The inter-comparisons .. showS´

Corrected in the revised version.

REF#2

I would like to thank the authors for considering and addressing all my comments and for the large effort they have undertaken for improving the manuscript. Many of them have been taken, which I acknowledge. Very valuable are, for example, the equilibrium index and the absorption correction.

I noted, however, that my major recommendations on the content of the manuscript have not been taken, e.g., to focus the manuscript only on the description of the dataset and present a proper validation of the whole dataset and do not present partial studies on a limited comparison and on the O3 behaviour but leave them for further studies.

I made those comments from the perspective of an eventual community user and for the potential of the paper for being useful and citable. With that option the message would be "here there is a new O3 middle atmosphere dataset". As it is written, however, the message is just that a dataset will be available in the future.

As for the comparison with other datasets and the discussion on the O3 behaviour, they are partial and incomplete studies (e.g., only some months, or the latitudinal dependence shown only for one month) and therefore the conclusions reached might not be fully valid. That is, as a user, I would not fully trust them. As an example, the authors show, as a typical example of the good agreement between IRI with OS and SMR, one orbit and one profile, from the many millions(?) available. Again, as a user, I would like to see a statistical study. That single profile in the bottom panel of Fig. 11 would tell me little.

It seems the authors would like to publish two papers on this dataset. One, this, with the message that there will be available a dataset and, another, where the proper dataset will be presented with its validation. I think that is also a fair view, although I do share. The same applies to the comparison and O3 behaviour, I would leave them out but if the authors prefer to include them, I would not object. Therefore, I am ready to accept the publication of this work in ATM with its current major contents.

I am giving below a couple of comments (the most important) on the revised manuscript and minor suggestions (mainly editing).

We thank the referee for their time and advice. We agree that a full analysis of the entire dataset would be the most useful for the users however that is not possible at this time - since only a small part of the dataset has yet been processed and the computational effort to process the remaining part is considerable. This paper is intended to present the potential of this hitherto unexploited dataset and its potential shortcomings.

1) Error budget. Overall, I am satisfied with the analysis. However, it seems there is a mixed up between random (also usually called instrumental, noise or precision) and systematic errors. Something which I understood nearly the end of the manuscript but needs to be clarified. For example, in legend of Fig. 1, we read "total error". This clearly does not include the systematic error discussed later on (Table 1). Is this total "calibration" error? or instrument (noise) error? Or a contribution to the total systematic error?
See also comment below on:
- the comment on l. 245.
- Legend of Fig. 4
- in l. 397 it is clearly written: "... has a "precision" of around 5-20\% based on the retrieval noise estimate". This should be stated from the beginning.
We have noted either a systematic or random error accordingly in the manuscript. See details below.

2) Lowermost altitude coverage. In some places it is written that the data start at 40 km but in others, e.g. lines 392-393, (and mainly in the figures) the lowermost height is 50 km. I would recommend to state as lower altitude 50 km, particularly in Table 2. The reason is that according to the figure shown in the response (absorption factor tables), there is already a considerable absorption at 40-45 km, between 40 and 60\% absorption. What I am surprised is that even with such absorption the AKs still show a high sensitivity, MR near 0.8.
We have revised the text regarding the lowermost height - see below.
However just to be clear the MR of 0.8 is under the assumption that the absorption correction is accurate. As long as the signal to noise ratio is sufficient, there will be a high sensitivity of the measurement.

Other comments:

Abstract. "completely new" -> "new"?
Removed in the revised version.

Abstract, l. 9. I would use "feasibility" instead of "performance". The latter implies a quantification (e.g. the performance of an instrument), which has been done only partially.
Corrected in the revised version.

Abstract, l. 13. Sentence: "We find that IRI appears to have a positive bias of up to 25\% below 75 km, and up to 50\% in some regions above." This is an example of my general comment above (to justify my view). This sentence is a typical example of characterisation of an available database. It may well be that you try to understand and correct this bias in the future so the actual database will not have such a bias. In that case this sentence is meaningless. Or would you process the whole dataset with that known bias?
The analysis is based on the data available at this point.
We have added the following sentence to the abstract "If the origin of the bias can be identified before processing the entire dataset, this will be corrected and noted in the dataset description"

l. 13 " ... data set about the overall atmospheric distribution of ozone." Do you mean the whole atmosphere? Between 50 and 100 km?
Added 'between 50 and 100 km' in the revised version

l. 42, equilibrium -> equilibrium,
Corrected in the revised version.

l. 66, change "novel technique" to "novel treatment" or "novel approach", as written later on. "Technique" embraces the whole inversion.
Corrected in the revised version.

l. 83 "venerable". This is a question of taste, very personal. I would avoid religious terms in scientific papers. You can use other adjetives as "very productive", useful, fruitful, etc. Also later on you sometimes use "believe" which I would change to "think".
Corrected in the revised version.

p. 7 Legend of Fig 1. (already mentioned above). "total error". This clearly does not include the systematic error discussed later on (Table 1). Is this total "calibration" error? or total instrument (noise) error? Or a contribution to the total systematic error? It should be clarified.
This refers to the total calibration random error. It is clarified in the caption in the revised version.

l. 186-7. "... and it changes with the atmospheric temperature at the tangent point.". In this case Phi should be inside the integral in Eq. 5.
Corrected in the revised version.

l. 190. suggested writing: " ... the atmospheric layer at the line-of-sight is optically thick."
Corrected in the revised version.

l. 194. "... is used for temperature and pressure." and I believe also for "O2 vmr".
Corrected in the revised version.

P. 8 legend of Fig. 2. I suggest to change "error size" to just "error" or "error values" or "error ratio or percentage". One possibility is to multiply it in this figure by 100 and give it in %.
Corrected in the revised version.

p. 10. Fig. 3. (Already mentioned above). If the absorption is so strong at tangent paths above 50 km (see fig. in the reply to the referees), how can the sensitivity be so large at 40 km?
Minor: label box partially overlap/hide the data.
The label box is moved outside of the plot in the revised version.
The high MR at 40 km is under the assumption that the absorption correction is accurate. As long as the signal to noise ratio is sufficient, there will be a high sensitivity of the measurement.

l. 245. Here it also needs to clarify which is this error. I believe it is the random (instrumental or noise error). Note that these values contrast with the upper limits listed in Table 1. So they need to be clarified.
This is the random error calculated from the inversion, Sm in Eq. 14. It is clarified in the text in the revised version.

l. 249. zic-zac -> zig-zag? or zigzag?
Corrected in the revised version.

Fig. 4. Again clarify which is this error?
Replaced 'error size' with 'random error'  in the revised version.

Fig. 7 (p. 16). Legend. I would change "naive" to "simple" or similar. "naive" is relative to persons, ideas, not to quantifications. Change it also in the text.
Corrected in the revised version.

ll. 392-393. " the lowest 10km grids in the retrieval are filtered out to avoid biases due to the possible edge effect." So, at the end, the lowermost retrieved altitude is 50 km? (Change it in Table 2).
Corrected to 50 km in Table 2 in the revised version.

l. 397. "... has a precision of around 5-20\% based on the retrieval noise estimate" OK. Now it is clear. This should be stated much earlier in the manuscript.
In the beginning of Sect. 2, the following sentences have been edited. 'The theoretical background, the implementation details and the intermediate results with their estimated random errors can be found in the corresponding subsections. At the end of this section, data availability at 80 km and the estimated systematic error sources are discussed.'

l. 403. OK. Now it is clear the difference between the precision (random error or noise) and the systematic error sources. Make this clear from the beginning, including the abstract.
It is clarified in the abstract and the beginning of Sect. 2 in the revised version.

p. 19, Table 1. Which the the precision or bias of the pointing?
Change "estimated error sizes" by "estimated errors"
Footnote. a), "... sensitivity of each of the parameter" -> " ... sensitivity of each parameter.
Corrected in the revised version.

p. 20, Table 2.
"Retrieval uncertainty" -> random error, precision, noise error (use one of these terms).
40-100 km -> 50-100 km
Corrected in the revised version.

ll. 413-414. I do not understand this sentence. The retrieved quantity is always number density. However, if you measured pressure and temperature simultaneously you can translate it, without lack of accuracy, to VMR, which is a better unit for understanding the chemical and physical processes behind, and do not show the large variation of several orders of magnitude.
We acknowledge that VMR is a better unit in some cases. However, OSIRIS does not have simultaneous measurements of temperature and pressure, thus the unit conversion involves external data such as MSIS, which ultimately may introduce yet another error source in the presented data in VMR unit.

l. 450 KIT-IMK -> KIT-IMK and IAA-CSIC

Corrected in the revised version.

l. 454. I suggest to add this reference for the MIPAS p-T data used: Garcia-Comas, M., Funke, B., Gardini, A., López-Puertas, M., Jurado-Navarro, A., von Clarmann, T., Stiller, G. P., Kiefer, M., Boone, C. D., Leblanc, T., Marshall, B. T., Schwartz, M. J. and Sheese, P. E.: MIPAS temperature from the stratosphere to the lower thermosphere: Comparison of vM21 with ACE-FTS, MLS, OSIRIS, SABER, SOFIE and lidar measurements, Atmos. Meas. Tech., 7(11), 3633–3651, 2014.

Added in the revised version.

Fig. 11. Legend, line 2. "(above 40 km)". It seems it is above 50 km (consistent with the whole manuscript).

Corrected in the revised version.

l. 463. "IRI ozone covers the altitude range from 50 to 100km as ..." This seems to be correct and more likely. Harmonise the rest of the manuscript with this value.

In part where we describe the analysis of resulting O3 data (Sect. 3),  we have corrected it to use 50 km instead of 40 km as the lower limit of the available O3 profiles. However,  in Sect. 2, where we describe the implementation methods, we select measurement vectors from 40 km without any filtering applied. Thus it remains as 40 km in the text and figures in Sect. 2.

l. 468. "... the background density included in the SMR product". Is the density measured by SMR? Clarify and mention the ultimate source used.

Clarified in the line as well as in Sect. 3.1.2.

ll. 471-472. As mentioned in the general comment above, this coincidence (lower panel) is fine. However, if one would like to have a solid consistency between the three datasets it should not rely on one single profile from ~millions. That is, as a user I would not trust it unless it is done on a statistically meaningful sample.

We agree with the referee that a single comparison is not convincing. We have however reviewed many (given hundreds not millions) and a note to this effect has been added to the text.

"While this is a single comparison, our general conclusion is that this holds for the majority of the profiles that we have inspected."

As already said earlier, a complete statistical analysis of the comparison results will be presented at a next stage, when the fully processed IRI data set will be available.

l. 474. believe > think
Corrected in the revised version.

l. 481. hemisphereS
Corrected in the revised version.

l. 493. "We are going to look at the ..." -> We discuss the ...
Corrected in the revised version.

l. 495. I think the study would be more solid if based on several months (not just one) but then this is probably "beyond the scope of the paper". This if the reason I argue in my general comment to focus the paper on one aspect and leave the other for other studies.
We have actually reviewed an entire year of data with similar conclusions for all months. We chose to present July since the overlap and geographical coverage with the other instruments was better.
The following sentence has been added to the beginning of Sect. 3.4:
'Similar conclusions can be drawn from the comparison of the other months of the year (not shown here).'

Fig. 12. Legend. Monthly mean DAYTIME ozone number density ...
Corrected in the revised version.

Fig. 13, legend, line 2. Again mention 50 km as the uppermost altitude. OK. consistent with everything else. How was performed the merging between the 2 datasets at 50-60 km?
The data have not been merged: the overlap region can be seen on close inspection of the contour lines. The colour however tends to hide this.

l. 521. "...where the differences are bigger, up to -70\%." This is correct but maybe you should warn that O3 vmr at 80 km is very small, which exploits the relative difference. What is the absolute diff in VMR?
The absolute difference in VMR is indeed very small at 80 km (up to 0.1ppm as shown in the figure below). The following sentence is added to the revised version.
'Note that the biggest relative differences observed are at the lowest ozone concentration.

[Figure]

I. 527. ... maybe be THE main reasonS ...
Corrected in the revised version.

Fig. 14. Left upper panel: The legend box hide some of the data
Legend, line 3. Specify which type of error.
Corrected in the revised version.

ll. 554-555 "...according to how much the measurement conditions
diverge from ..." -> ... according to the divergence from ...
Corrected in the revised version.

I. 563. "The comparison also demonstrates the advantage of the high
sampling rate of IRI." Probably the major advantage would be to reduce
the precision of the mean values when averaging large samples?
We agree that could be one advantage.
The following has been added to the manuscript.

[revised manuscript text omitted]